# Spatiotemporal Source Apportionment of Ozone Pollution over the Greater Bay Area

Yiang Chen [1], Xingcheng Lu [2][*] Jimmy C.H. Fung [1, 3]

1 Division of Environment and Sustainability, The Hong Kong University of Science and Technology, Clear Water Bay, Kowloon, Hong Kong SAR, China

2 Department of Geography and Resource Management, The Chinese University of Hong Kong, Sha Tin, New Territory, Hong Kong SAR, China

3 Department of Mathematics, The Hong Kong University of Science and Technology, Clear Water Bay, Kowloon, Hong Kong SAR, China

*Correspondence to*: Xingcheng Lu (xingchenglu2011@gmail.com)

**Abstract.** It has been found that ozone ($O_3$) pollution episodic cases are prone to appear when the Greater Bay Area (GBA) is under the control of typhoons and sub-tropical high-pressure systems in summer. To prevent these pollutions effectively and efficiently, it's essential to understand the contribution of $O_3$ precursors emitted from different periods and areas under these unfavorable weather conditions. In this study, we further extended the Ozone Source Apportionment Technology (OSAT) from the Comprehensive Air Quality Model with Extensions (CAMx) model to include the function of tracking the emission periods of $O_3$ precursors. Subsequently, the updated OSAT module was applied to investigate the spatial-temporal contribution of precursor emissions to the $O_3$ concentration over the GBA in July and August 2016, when several $O_3$ episodic cases appeared in this period. Overall, the emissions within the GBA, from other regions of Guangdong province (GDo), and the neighbouring provinces were the three major contributors, accounting for 23%, 15%, and 17% of the monthly average $O_3$ concentration, respectively. More than 70% of $O_3$ in the current day was mainly formed from the pollutants emitted within 3 days and the same day's emission contributed approximately 30%. During the $O_3$ episodes, when the typhoon approached, more pollutants emitted 2-3 days ago from the GDo and adjacent provinces were transported to the GBA, leading to an increase in $O_3$ concentrations within this region. Under the persistent influence of northerly wind, the pollutants originating from eastern China earlier than 2 days ago can also show a noticeable impact on the $O_3$ over the GBA in the present day, accounting for approximately 12%. On the other hand, the $O_3$ pollution was primarily attributed to the local emission within 2 days when the GBA was mainly under the influence of the sub-tropical high-pressure systems. These results indicate the necessity to consider the influence of meteorological conditions in implementing the control measures. Meanwhile, analogous relationships between source area/time and receptor were derived by the zero-out method, supporting the validity of the updated OSAT module. Our approach and findings could offer more spatial-temporal information about the sources of $O_3$ pollutions, which could aid in the development of effective and timely control policies.

## 1. Introduction

As one of the major air pollutants, ozone ($O_3$) is a secondary pollutant formed by the photochemical reactions of nitrogen oxides ($NO_x$) and volatile organic compounds (VOCs) in the presence of solar radiation. Surface $O_3$ has detrimental effects on human health, such as causing respiratory and cardiovascular problems (Maji et al., 2019; Yin et al., 2017). It could also lead to the reduction of crop yield and the damage of vegetation (Gong et al., 2021; Wang et al., 2022c). With the implementation of a series of control policies in China since 2013, the concentrations of other air pollutants, including particulate matter with aerodynamic diameters less than 2.5μm ($PM_{2.5}$), $NO_x$, and sulfur dioxide ($SO_2$), have gradually decreased. In contrast, due to the large reduction of $NO_x$ emission and limited control of VOCs emission in the early stage of the control period (Liu et al., 2023), the $O_3$ concentration still continuously increased and has become the primary air pollutant across China. The Greater Bay Area (GBA), including nine cities in the Pearl River Delta (PRD) region, Hong Kong (HK), and Macau Special Administrative Regions (SAR), is one of the most developed agglomerations in China and also faces the heavy $O_3$ pollution problem. Based on the analysis of surface monitor observation, Cao et al. (2024) and Feng et al. (2023) revealed

an overall upward trend in the maximum daily 8-h average (MDA8) $O_3$ in the PRD region and HK, with an
increase of 1.11 and 0.22 ppbv/year from 2013 to 2019 and from 2011 to 2022, respectively.
The formation of $O_3$ is closely related to the sources of its precursors, and much effort has been devoted to
investigating the source region and source category of $O_3$ in the GBA using different methods (Liu et al., 2020a).
He et al. (2019) applied the positive matrix factorization (PMF) method to resolve the anthropogenic sources of
VOCs. Combining a photochemical box model with the master chemical mechanism (PBM-MCM), they found
that vehicular was the most significant source of the $O_3$ formation, followed by biomass burning and solvent usage.
Li et al. (2012) applied the CAMx-OSAT numerical model to track the source contribution to $O_3$ in the GBA
region and found that elevated local and regional contributions were dominant during the $O_3$ episodes. Yang et al.
(2019b) applied the NAQPMS model with an online source apportionment module to explore the sources of $O_3$
in different seasons in the PRD region. Their results showed that the mobile was the largest contributor, followed
by industry. Fang et al. (2021) used multi-modelling source apportionments to quantify the source impact on $O_3$
in the PRD region. The on-road mobile and industrial process were found to be two major contribution sectors.
Integrating satellite data and sensitivity model simulations, Wang et al. (2022a) found that enhanced biogenic
emission and cross-regional transport due to approaching typhoons were significant factors leading to ozone
pollution in the PRD and Yangtze River Delta (YRD) regions. In addition to the source region and category, the
emitting time of pollutants is also an important perspective that needs a better understanding for effective and
efficient control policymaking. Several studies have attempted to evaluate this temporal perspective (Xie et al.,
2021; Ying et al., 2021). Xie et al. (2023) analysed the age evolution of $PM_{2.5}$ during a haze event in eastern China.
It showed that during the regional transport stage, more aged particles from the North China Plain (NCP) were
transported to the downwind YRD region, leading to a sharp increase in the average age of different components
of $PM_{2.5}$ in the YRD. Chen et al. (2022c) investigated the temporal contributions of emissions to the concentration
of $PM_{2.5}$ in the PRD region and found that pollutants emitted 2 days earlier were trapped within the PRD region
due to the weak wind during the episodic pollution. However, these studies mainly focused on the $PM_{2.5}$ and the
temporal contribution of sources to the $O_3$ in the GBA region still remains unclear.
In addition to emission,  meteorological conditions, another key factor that can affect the transportation,
production, and destruction of $O_3$ and its precursors, have also received much attention and have been extensively
studied (Lu et al., 2019; Wang et al., 2017; 2022b). The long/short-term effects of meteorological changes on
ozone concentrations have been investigated through various methods, such as statistical analysis of observations
and numerical modelling (Yang et al., 2019a; Xu et al., 2023a; Zheng et al., 2023). Liu and Wang (2020b)
conducted sensitivity simulations by the CMAQ model to evaluate the contribution of variations in weather
conditions to summer $O_3$ levels from 2013-2017. Their results showed that the meteorological conditions were
more conducive to ozone formation from 2014 to 2016 than in 2013, leading to an increase of more than 10 ppbv
in MDA8 $O_3$ in Guangzhou. Different objective and subjective classification technologies have been applied to
summarize the impacts of unfavorable weather patterns on $O_3$ pollution (Han et al., 2020; Chen et al., 2022b; Cao
et al., 2023). Gao et al. (2018) summarized the common synoptic patterns in the Guangdong province that $O_3$
pollution always occurred and concluded that the sub-tropical high-pressure system and typhoons are two major
patterns accounting for more than 60% of cases in the PRD regions during 2014 - 2016. The major influencing
factors and the dominant physical and chemical processes were also identified and analyzed (Gong et al., 2022;
Zeren et al., 2022; Wu et al., 2023). Ouyang et al. (2022) analysed the impact of a subtropical high and a typhoon
on ozone pollution in the PRD region and found that low relative humidity, high boundary layer height, weak
northerly surface wind, and strong downdrafts were the main meteorological factors contributing to the pollution.
Deng et al. (2019) illustrated that the actinic flux was the important cause of the co-occurrence of high ozone and
aerosol pollution under the control of typhoon periphery. Li et al. (2022) also investigated the impact of peripheral
circulation characteristics of typhoons and found that the chemical formation and vertical mixing effects were two
major contributors to the enhancement of $O_3$ levels, while the advection showed negative values. Qu et al. (2021)
analysed the typhoon-induced and non-typhoon $O_3$ events in the PRD region and revealed that under the influence
of typhoons, the contributions from the transport processes and sources outside the PRD increased. Usually, the
ozone events are attributed to changes in meteorological conditions rather than sudden increases in emission
intensity (Lin et al., 2019; Xu et al., 2023b). The changes in weather conditions will affect the time-sensitivity of
emitted pollutants and lead to different types of $O_3$ pollution, such as long-range transport of aged pollutants or
accumulation of local fresh pollutants. Hence, it is of great importance to clarify the impact of the pollutants from
different source areas and emitting periods on the $O_3$ pollution under different weather conditions in the GBA.
In this study, the CAMx-OSAT model was extended and used to track the temporal contribution of pollutants to
the $O_3$ pollutions over the GBA under the impact of typhoons and sub-tropical high pressure during July and
August in 2016, the two most important weather systems that influence O₃ pollutions over the GBA. The rest of
this paper is organized as follows. The temporal source apportionment (TSA) method, the configuration of
experiments, and the ozone episodes are introduced in section 2. The spatial-temporal source apportionment
results and zero-out simulation results are shown and discussed in section 3. The major conclusions are
summarised in section 4.

## 2. Methodology and Data

### 2.1 Temporal Source Apportionment Method

Previously, we have successfully implemented the $PM_{2.5}$ temporal source apportionment method in the CAMx
model and applied it to investigate the temporal influence of emissions on $PM_{2.5}$ in the GBA (Chen et al., 2022c).
Here, we further extend this method to track the temporal contribution of emissions to the precursors and the
formation of O₃. Similar to the OSAT method, the input data used in the TSA method developed in this work
include the source area map and hourly emission data. The source area map assigns each model grid cell to one
of the specific source regions. The hourly emission data is the same as the one used in the normal CAMx model
simulation without turning on the source apportionment module. The basic mechanism of the TSA method is to
track the contribution of pollutants from different emitting periods using a set of tracers. In the TSA method (Fig.
1), the *Precursor Tracer Day-x* was used to track the precursors emitted from *x* days ago. The *O₃ Tracer Day-x*
was used to track the O₃ formed from the precursors emitted from corresponding *x* days ago (namely *Precursor*
*Tracer Day-x*). The tracers in *Day-x* can be set into different finer periods (e.g., every 1 hour, 6 hours, 24 hours)
as required. The total number of tracers will be decided according to the entire tracking period and the minimum
tracking period per tracer. For instance, if the entire tracking period is 5 days and the minimum tracking period
per tracer is every 6 hours, the total number of tracers will be 20. In each time step, the tracers go through all the
processes, including emission, transport, diffusion, and chemical reactions, sequentially, as in the normal CAMx
model simulation. Therefore, the precursors and O₃ tracers that tracked different periods are calculated
simultaneously. When the pollutants emitted from the sources, they will be assigned to the *Precursor Tracer* in
Day-0, while the *Precursor Tracers* that tracked other periods and the *O₃ Tracers* remain unchanged. The data
transfer between tracers (e.g., Day-1 to Day-2, and Day-0 to Day-1, dash arrow in Figure 1) will be conducted
once after one day's simulation. As shown in Figure 1, during each day's simulation, the contribution of the present
day's emission is consistently tracked by the Day-0 tracers. After completing the current day's simulation and
before starting the next day's simulation, each tracer *Day-x*'s value transfers to the corresponding tracer *Day-(x+1)*,
which represents one day earlier than *Day-x*, following the specified sequence. For example, beginning from the
penultimate tracer, namely values in Day-3 transfer and add into Day-4, then the values in Day-2 transfer to Day-
3, followed by Day-1 to Day-2, and lastly Day-0 to Day-1 (Dash arrow in Figure 1). Here, the value in Day-3
tracer will be added into the last tracer (Day-4) because the last tracer represents the total contribution of pollutants
emitted earlier than 3 days ago. Same as the OSAT method, the TSA method also utilizes the photochemical
indicator, namely, the ratio of the production rate of hydrogen peroxide ($H_2O_2$) and nitric acid ($HNO_3$), to
determine the sensitivity of O₃ formation. When the O₃ formation is classified as NOx-limited (VOC-limited), the
contributions are distributed to the NOx (VOCs) sources emitted at different periods, based on the proportion of
their emissions to the total NOx (VOCs) emissions. More details of this method can be found in Chen et al. (2022c).

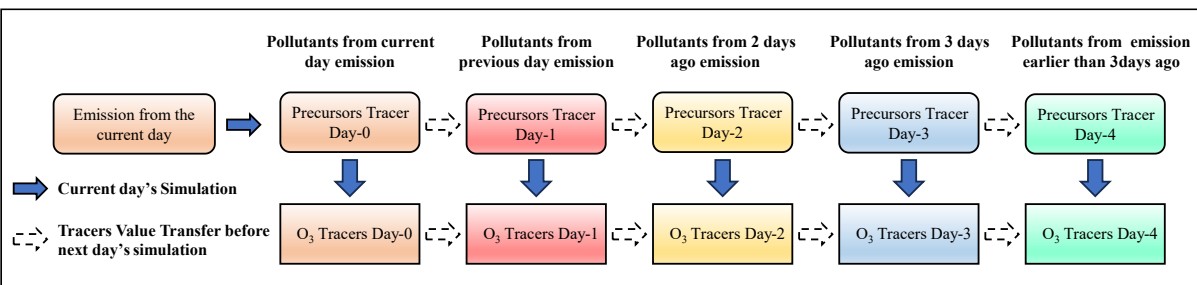


Figure 1. Schematic diagram of temporal source apportionment (colors represent the pollutants released or formed
by emissions on different days).

## 2.2 Model Configuration and Evaluation

The Weather Research and Forecasting (WRFv3.9) model was applied for meteorological field simulation. The initial and boundary condition for the WRF model was gained from the Final Operational Global Analysis data (FNL). The CAMx v7.1 was used to simulate the spatial-temporal variation of air pollutants. The initial and boundary condition for the CAMx model was provided by the Model for Ozone and Related chemical Tracers, version 4 (MOZART-4). Regarding the emission, a highly resolved emission inventory provided by the Hong Kong Environmental Protection Department (HKEPD) was used for the GBA region, and the Multi-resolution Emission Inventory for China (MEIC, Li et al., 2017) developed by the Tsinghua University was applied for the area outside the GBA region. The biogenic emission for the entire domain was calculated by the Model of Emissions of Gases and Aerosols from Nature (MEGAN version 3.1). The CB05 gas phase chemistry, the ISORROPIA inorganic aerosol scheme, and the SOAP secondary organic aerosol scheme were used in the simulation. This model system has been applied to analyse the source of $O_3$, $NO_x$, and $PM_{2.5}$ in the GBA region in previous studies (Lu et al., 2016; Chen et al., 2022a; Chen et al., 2022c). More configuration of this model system can refer to the work of Lu et al. (2016).

The three-nested simulation domain of the WRF-CAMx model was shown in Figure S1. The resolution of three domains was 27km, 9km, and 3km, respectively. For the source apportionment experiments, the simulation domain was divided into 12 source regions as shown in Figure 2, including North China Plain (*NCP*), eastern China(*EC*), southern western China (*SWC*), other regions of inland China (*Other 2*), ocean and other countries (*Other 1*), neighbouring provinces around Guangdong province (*Neighbor*), Other region within Guangdong province but outside the GBA(*GDo*), different sub-regions within the GBA: Guangzhou and Foshan(*GF*), Shenzhen and Dongguan(*SD*), Hong Kong (*HK*), Zhuhai, Zhongshan and Jiangmen (*ZZJ*), Zhaoqing and Huizhou(*GBAo*). The cities within the GBA were separated into different sub-regions mainly based on administrated boundaries and their geographical location, same as the work of Chen et al. (2022c). The sub-regions mainly consist of neighboring cities. Zhaoqing and Huizhou, located at the northwestern and northeastern corners, respectively, were categorized into one group since they have a relatively lower emission density than other cities. Previous studies indicated that the air pollutants in Hong Kong were usually more influenced by long-range transport from regions outside the GBA, in contrast to the other cities in the GBA (Li et al., 2012; Chen et al., 2022a; Chen et al., 2022c). Hence, Hong Kong city is treated as a separate entity. The contribution of initial and D1 boundary conditions were also treated as two sources. In the following analysis, for the $O_3$ concentrations in the target area over the GBA, the influence of pollutants emitted within the target area is treated as the local contribution, and the influence of pollutants originating from the other areas within the GBA region is treated as the regional contribution. The source tracking time period is 5 days. Day-0, Day-1, Day-2, Day-3 represent the pollutants emitted within the present day, the previous day, two days ago, and three days ago, respectively. Day-4 represents the total contribution of pollutants emitted earlier than three days ago. The simulation period is July and August 2016, and the model was spin-up for 7 days to reduce the influence of initial condition.

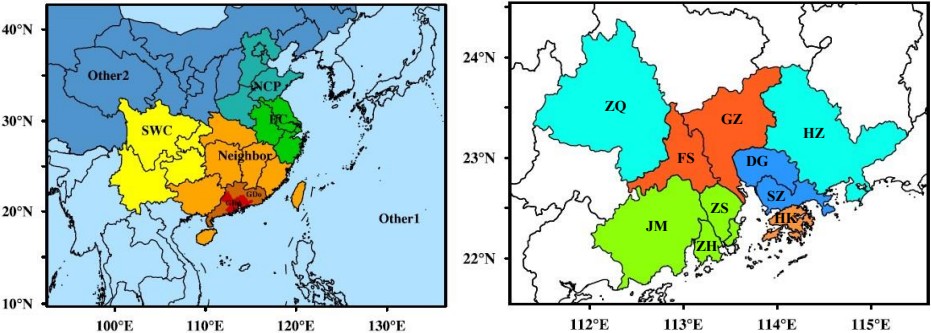

Figure 2. The configuration of source areas in the source apportionment experiments (One color represents one source area. The GBA source were divided into five source areas. *Other 1* represents ocean and other countries. *Other 2* represents other area within the mainland China in the simulation domain.)

The performance of simulated hourly 2-m temperature, 10-m wind speed, and $O_3$ concentration were evaluated and shown in Table S1. Here, the statistical metrics, including mean bias (MB), normalized mean bias (NMB), index of agreement (IOA), and root mean square error (RMSE), were used for model performance evaluation. The mathematical formulas for these metrics can be found in Table S6. The recommended values suggested by

Emery et al. (2001) and EPA (2007) were used as benchmarks and shown in the brackets in Table S1. The
temperature is a little overestimated with a MB of 0.33, while the wind speed is underestimated with a MB of -
0.45. The IOA is 0.82 and 0.70 for temperature and wind speed, respectively. The MBs and IOAs both fulfill the
criteria. But the RMSE shows a little higher than the value of criteria. Regarding the $O_3$, the IOA reaches 0.81.
The small positive MB indicates that the model slightly overestimates the $O_3$ concentration. The NMB is 0.13,
which also meets the criteria. The time series comparison (Fig. S2) of average $O_3$ concentration in Guangzhou,
Hong Kong and Zhuhai illustrates that the model can well catch and reproduce the variation trend of $O_3$
concentration in GBA, although there are a few differences between the simulated and measured concentration
for some peaks, like the period between 25 July and 31 July in Guangzhou. Overall, the performance of model
simulation is comparable to the other studies in this region (Li et al. 2022; Yang and Zhao, 2023). Therefore, the
simulation result is reasonable and can be further used for source analysis.
**2.3 Ozone Episodes**
There were several $O_3$ episodes that occurred during the simulation period. Here, the maximum daily 8-h average
(MDA8) $O_3$ concentration over the GBA was calculated using the observation data from the surface monitors
stations (Fig. 3). The $O_3$ observations were obtained from the China National Environmental Monitoring Centre
(CNEMC) and the HKEPD. Here, pollution days were identified when the average MDA8 $O_3$ observations
concentrations over the GBA exceeded 80ppb (Wang et al., 2022d). To better capture the evolution of the $O_3$
pollution, based on the characteristics of concentration variation, the days preceding and following the $O_3$
pollution days were also included in the analysis and the whole period was considered as an $O_3$ episode. The first
$O_3$ pollution occurred between the 7[th] and 10[th] of July (Ep1). During this period, the GBA region was initially
controlled by the sub-tropical high-pressure system. When the typhoon north-westerly moved from the east sea
area of the Philippines towards Taiwan province, the GBA was located in the peripheral subsidence region. After
the typhoon made landfall, the high-pressure situation in the GBA was relieved and the $O_3$ concentration decreased.
There were another two $O_3$ episodes between 24 July and 1[st] August. The GBA was mainly influenced by the sub-
tropical high-pressure system during 24[th]-26[th] July (Ep 2), while the synoptic condition of the GBA between 30[th]
July-1[st] August (Ep3) was similar to that of Ep1. During the Ep3 , another typhoon moved north-westerly from
the east sea area of the Philippines and influenced the GBA region. It was found that this type of typhoon
movement path was often accompanied by the occurrences of $O_3$ pollution in the GBA (Wang et al., 2022a). In
late August, under the joint influence of the subtropical high-pressure system and the typhoon, the $O_3$ over the
GBA maintained a high concentration level between the 21[st] -31[st] of August (Ep4). Unlike previous two typhoons,
this typhoon moved southerly from the sea areas south of Japan and stayed near the sea areas east of Taiwan
province. The typhoon moved northwards after 27[th] August, and northerly winds prevailed in the GBA. Hence,
we conducted the simulation of $O_3$ concentration in the GBA during July-August 2016 and analysed the
spatiotemporal contributions of emissions in these episodic cases.

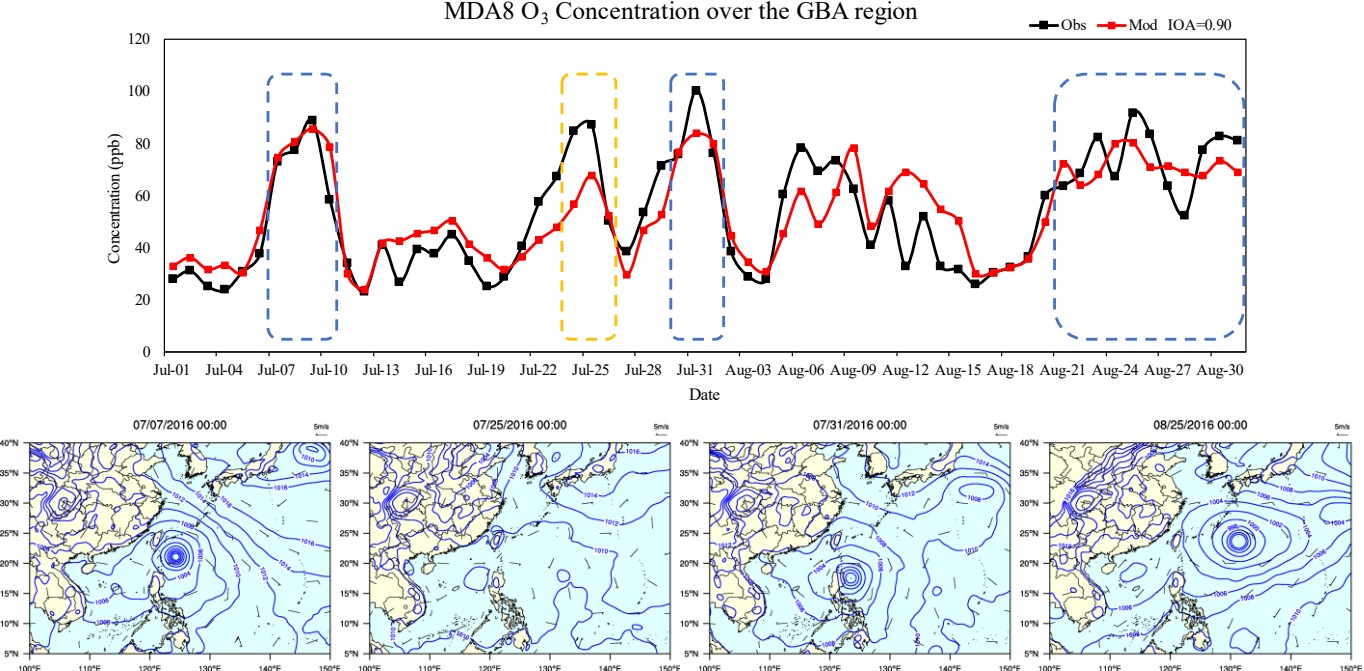

Figure 3. The time-series of the observed and simulated MDA8 $O_3$ concentration over the GBA during July-August 2016 and the synoptic patterns during the $O_3$ episodes. (Blue box: typhoon case; Yellow box: sub-tropical high-pressure case. The $O_3$ observations were obtained from the CNEMC and the HKEPD. The synoptic patterns were plotted using the ERA5 reanalysis data)

## 3. Result and Discussion

### 3.1 Source Area Contributions

The contribution of different source areas to the average hourly $O_3$ concentration in the GBA region is shown in Table 1. Here, the contribution from initial and boundary conditions were treated as background contribution. Regarding the monthly average $O_3$ concentration over the GBA region, the emission within the GBA can contribute about 23%. The pollutants from other regions within Guangdong Province (GDo) and neighbouring provinces also had large contribution, accounting for approximately 15% and 17%, respectively. Under the influence of prevailing south winds in the summertime, the contribution from ocean and other countries can also account for about 20%. As some studies suggested that $O_3$ originating from foreign countries is quite limited (Sahu et al., 2021), the main contributor of this source is likely to be marine ship emissions from ocean. The pollutants from other source regions had limited effect on the $O_3$ in the GBA.

The monthly average source area contribution to four sub-regions within the GBA region can be found in Table S2. Results show that the local emission had a significant influence on $O_3$ in the GF and SD regions, accounting for 17% of $O_3$ but its impact was lower than 10% on $O_3$ in the ZZJ region and HK city. The contribution of GBA regional emissions (contributed by other GBA tagged regions) had a relatively larger impact on the monthly average $O_3$ concentration in the GF region than the other sub-regions. It's because of the prevailing southerly wind in summer, which resulted in a greater influence of the pollutants within the GBA region on $O_3$ in the GF area. The influences of pollutants from GDo and neighboring provinces on different subregions ranged from 25% to 31%. As coastal regions, the ZZJ region and HK city were also more affected by sources of ocean and other countries, which occupied about 24% and 27%, respectively.

Regarding the average hourly $O_3$ concentration over the GBA region in different episode periods, it can be found that, during the typhoon episodes (i.e., Ep1, Ep3 and Ep4), the contribution of non-local emission has increased. The typhoon paths were quite similar in the Ep1 and Ep3 episodes (Fig. S3). Results show that the total contribution of GDo and neighbouring provinces have increased and reached more than 50% for $O_3$ over the GBA in these two typhoon episodes. As shown in Figure S4, with the approaching of the typhoon, the wind speed

increased and the average wind direction over the GBA changed from south to north. Therefore, more pollutants from the surrounding provinces were transported to the GBA. Considering the typical circulation patterns of the typhoon periphery (Figure. S4 and S6), it is inferred that more pollutants may come from Jiangxi, Fujian, and Hunan provinces. During the Ep1 and Ep3 episodes, the contribution of local emission in different sub-regions slightly decreased. With the change of the wind direction from south to north in these two periods, the influence of pollutants within the GBA to $O_3$ in the GF area decreased from 15% to 8%. The contribution of the GBA emission to the $O_3$ in other sub-regions increased, especially the ZZJ area and HK city. It is because of the change of wind direction, these two regions were located at the downwind area of the GF and SD regions, which are the emission hotspots within the GBA. At the same time, the contribution of source from ocean and other countries also decreased by approximately 10%. The contribution of emission from the GDo and neighboring provinces to $O_3$ concentration in GF, SD, ZZJ regions, and HK city increased by 27%, 21%, 32%, and 22%, respectively.

In another typhoon process (Ep4), where the typhoon's moving path differed from the other two typhoon cases, an increase in the contribution from GDo and neighbouring provinces was observed due to the persistent northerly winds. Furthermore, it was observed that pollutants from eastern China (EC) and North China Plain (NCP) could also influence the $O_3$ levels in the GBA, accounting for approximately 12%. Similar increases in the impact of emissions from the EC and NCP were also found in the four sub-regions.

In the Ep2, the GBA was mainly controlled by the sub-tropical high-pressure system, with prevailing southerly wind. However, the low wind speed was conducive to the accumulation of the pollutants. Hence, the local sources were the dominant contributors and accounted for about 44%, while the contribution from GDo and neighboring provinces decreased. For $O_3$ in the GF region, as discussed above, the $O_3$ in the GF region is more susceptible to emissions within the GBA under the prevailing southerly wind. Thus, not only the local contribution but also the GBA regional contribution largely increased in the GF region. The regional contribution is larger in the GF region, increasing from 15% to 33%. On the other hand, the main increase in other sub-regions was seen in the local contributions.

Table 1. Contribution of pollutants from different source areas to the average hourly $O_3$ concentration over the GBA in different cases.

| Case | GBA | GDo | Neighbor | Other 1 | EC | SWC | NCP | Other 2 | Background |
|---|---|---|---|---|---|---|---|---|---|
| Monthly | 23% | 15% | 17% | 20% | 3% | 1% | 1% | 1% | 20% |
| Ep1 | 18% | 21% | 35% | 10% | 3% | 0% | 0% | 0% | 13% |
| Ep2 | 44% | 11% | 7% | 27% | 0% | 0% | 0% | 0% | 11% |
| Ep3 | 19% | 34% | 25% | 9% | 3% | 0% | 1% | 1% | 9% |
| Ep4 | 20% | 16% | 18% | 15% | 8% | 1% | 4% | 3% | 14% |

* Here, *GDo* represents areas outside the GBA but within Guangdong province. *Neighbor* represents the provinces around Guangdong province. *Other 1* represents ocean and other countries. *Other 2* represents other areas within the mainland China in the simulation domain. *Background* represents the contribution of initial and boundary conditions.

### 3.2 Emission Period Contributions

The contribution of pollutants emitted from different time periods to the average hourly $O_3$ concentration in the GBA and its sub-regions is shown Figure 4 and Table S3. The background contribution was not considered in the temporal source contribution analysis. This is because the background contribution is primarily derived from boundary conditions, and its temporal contribution is calculated based on the time when the pollutants are transported into D1, rather than the actual emission time.

Overall, under the general monthly condition, the emissions within 3 days (namely from Day-0 to Day-2) account for approximately 73% of the monthly average $O_3$ concentration within the GBA. The largest proportion of $O_3$, around 31%, was formed from the current day's emission (Day-0) and the contribution of pollutants from earlier emission periods decreased as time elapsed. For the monthly average $O_3$ in different sub-regions, more $O_3$ in the GF and SD regions was formed from the emission from Day-0, which contributed about 37% and 36%, respectively. The contribution of emissions from Day-1 decreased to about 23% in these two regions. The contribution of Day-0 and Day-1 emissions was relatively small but stable for the HK city and ZZJ region, which

accounted for around 25% and 27%, respectively. The influence of pollutants emitted earlier than 3 days ago (i.e.,
Day-4) was generally lower than 20%.
The situations are different during the pollution periods. The contribution of emission from the current days to the
average hourly $O_3$ over the GBA both decreased in the two typhoon cases with similar moving paths (Ep1 and
Ep3). However, the contribution of emissions from Day-1 to Day-3 increased 14% and 8%, respectively. And the
influence of pollutants emitted earlier than 3 days ago (Day-4) decreased 11% in Ep1 and remained almost
unchanged in Ep3. These findings indicate that these two ozone pollutions were caused by the accumulation of
pollutants within the current 3 days.
For another typhoon case (Ep4), the contribution from Day-0 decreased approximately by 11%, compared to the
monthly contribution over the GBA. At the same time, the influence of pollutants from earlier emitting periods
increased, especially for those emitted earlier than 3 days ago. It means that the $O_3$ pollution during this period
was a persistent pollution process. The major contributor should involve not only local emissions but also long-
range transport. Similar trends in temporal contribution variations were observed in different sub-regions, which
also illustrated that $O_3$ pollution is usually a regional problem.
For Ep2, the contribution of emissions from Day-0 increased approximately 18%, while the influence of emissions
from Day-1 to Day3 decreased about 18%. According to the source area contribution result, the source area of $O_3$
over GBA in Ep2 is mainly local sources. Therefore, the contribution of freshly emitted pollutants was larger. The
contribution of Day-4 emissions to the HK city and ZZJ region in Ep2 waslarger. It is probably because of the
prevailing south wind direction, which brought more airflow from the ocean. Compared with the emission of the
GF and SD regions, the HK city and ZZJ region have lower emission amounts. At the same time, HK city and
ZZJ region were located in the upwind region, and the pollutants from GBA would have a smaller influence on
the $O_3$ in these two regions. Hence, the amount of fresh pollutants was smaller and contributed similarly to Day-
4 emissions, which is an accumulated amount.

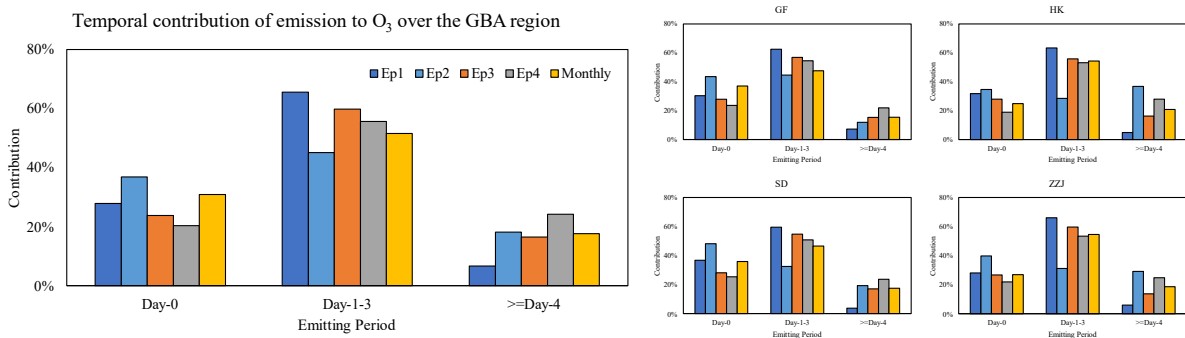

Figure 4. Contribution of pollutants from different emitting periods to the average hourly $O_3$ concentration over
the GBA in different cases.

### 3.3 Source Area-Time Contributions

To further clarify the relationship between sources and the $O_3$ concentration in target regions, the evolution of $O_3$
from various source areas and periods was analyzed. Figure 5 shows the time series of the contributions from
different source areas and precursor emission periods to the hourly average $O_3$ concentration in the GBA region.
Regarding the monthly average $O_3$ concentration over the GBA, the emission within the GBA was the major
contributor and generally had a larger effect on the current day. Under the control of southerly wind, as shown in
Figure 6, the pollutants emitted 1 day ago (Day-1) were gradually transported out of the GBA, and the influence
of the GBA's emission earlier than Day-1 diminished. Simultaneously, the pollutants of GDo and neiboring
provinces emitted 1 day ago began to have an impact on the $O_3$ in the GBA. However, due to the prevailing
southerly wind, the impact of aged pollutants from GDo and neighboring provinces on the $O_3$ in the GBA was
relatively low.
However, regarding the $O_3$ pollution between 7th and 10th July (Ep1), the major contributors changed. On 7th July,
the GBA was under the control of the subtropical high-pressure system, and the typhoon was located near the east
of Taiwan province. The weather condition was unfavourable for pollutants dispersion, and the $O_3$ sourced from
Day-1 emission within Guangdong provinces was trapped. The prevailing wind shifted to northerly wind, bringing
elder pollutants from neighboring provinces to the GBA. With the approach of the typhoon from 8[th] –10[th] July,
the stronger northwest wind speeded up the diffusion of pollutants from the GBA and deceased the local
contribution. However, it also transported more elder pollutants from the northern inland to the GBA. It can be
found that the emissions from GDo on the present day also had a significant contribution. At the same time, the
pollutants from the neighboring provinces dominated the emissions from Day-1 to Day-3. Moreover, the
pollutants emitted 2 days ago in the EC were also transported southward and affected the $O_3$ in the GBA on the
current day. Figure 7 shows the spatial distribution of the average source contribution during the Ep1 period.
Compared with the monthly average (Figure 6), it was found that the elder pollutants originating from the GBA
can be transported back and influence the $O_3$ concentration in the western part of the GBA during the Ep1 period.
This is because easterly winds blew over the GBA from 5[th]-6[th] July (Before Ep1, Figure S4). The pollutants emitted
within the GBA were transported to northwest inland. However, under the influence of northwest wind, they were
transported back to the GBA again. It can also be seen that the pollutants from the GDo 1 day ago were transported
downwind quickly, contributing to a high $O_3$ concentration over the Pearl River Estuary. According to the wind
pattern, they mainly came from the northern and western parts of the Guangdong province. Meanwhile, the
neighboring provinces' emissions from Day-1 to Day-3 were also transported to the GBA by the northwest wind,
continuously affecting the $O_3$ over this region.
For the Ep3 $O_3$ pollution process, results show that the pollutants from GDo and neighboring provinces were also
the major contributors. From 30[th]– 31[st] July, the GBA was under the control of high pressure, and weak north wind
prevailed in this region. Afterward, the approaching of the typhoon (1[st] August) further strengthened the cross-
regional transport of pollutants. The difference between Ep3 and Ep1 is that the emissions from GDo have a larger
proportion in the Day-1 and Day-2 emissions. Additionally, while pollutants from neighboring provinces and EC
in Day-4 emission only accounted for about 5ppb in Ep1, they can still contribute to about 10ppb in Ep3. The
possible reason is that northerly wind prevailed over Fujian, Jiangxi, and Hunan provinces during the whole Ep1
period (Figure S4). However, easterly wind still blew over these provinces during the earlier period of the Ep3
(30[th]– 31[st] July, Figure S6), which slowed the transport and influence of pollutants from the neighboring provinces.
Generally, the pathways of typhoons in the Ep1 and Ep3 episodes were quite similar, and the influence regions of
typhoon wind field mainly covered Guangdong and neighboring provinces. Therefore, the major source area and
source time were quite similar in these two cases. To prevent this type of $O_3$ pollution, earlier emission control (at
least 3 days ago) and collaboration with neighboring provinces will gain a better control result.
On the other hand, the situation is different for the Ep2 ozone pollution. Under the control of the high-pressure
system and weak southerly wind (Figure S5), the major contributors were mainly the pollutants from the GBA
and the ocean. Unlike the Ep1 and Ep3, the pollutant emitted within the GBA was still dominant in the contribution
of Day-1's emission. Under the influence of southerly wind, there was minimal migration of pollutants from north
inland regions to the GBA, and the local pollutants were gradually dispersing from the GBA. Thus, the pollutant
emitted earlier than 2 days ago (>=Day-2) had a smaller contribution. As shown in Figure 8, the overall diffusion
of pollutants within the Guangdong province was much slower during Ep2. The contribution of the GBA emissions
can still reach more than 10 ppb in the Day-1 emission. These results indicate that this pollution process was
mainly driven by the local pollutants within the current 2 days. Hence, emission control should focus on the local
sources, and 1-2 days in advance is more efficient.
For the last $O_3$ pollution process (Ep4), which occurred from the 21[st] to 25[th] August, eastern and southern China
were mainly controlled by the sub-tropical high-pressure system. Meanwhile, under the joint influence of
peripheral subsidence airflow of typhoon, the wind speed over this region was slow (Figure S7). The weak wind
not only trapped the $O_3$ formed from local emission but also the $O_3$ formed from cross-regional transported
pollutants. The pollutants from the GBA sources mainly dominated the Day-0 and Day-1 emission's contribution,
while Day-2 and Day-3 emissions mainly consisted of pollutants from GDo and neighboring provinces.
Subsequently, as the typhoon moved northward, the stronger northerly wind further broadened the source areas of
the $O_3$ in the GBA (Figure S7). The major contributor of Day-2 and earlier periods' emissions changed to pollutants
from the EC and NCP regions. The pollutants emitted earlier than 2 days ago from the EC had an important
contribution, which accounted for about 12%. Furthermore, the pollutants emitted 3 days ago from the NCP can
also have a noticeable impact on $O_3$ over the GBA from July 28[th]- 30[th], which can be up to 10%. Therefore, to
prevent the occurrence of this pollution, emission control measures should be implemented in a broader region
and continuously enforced, as this pollution episode lasted longer compared to the other three cases.
Figure S8 shows the time series of the contributions from different source areas and precursor emission periods
to the hourly average $O_3$ concentration in the GF region and HK city. GF region is located at the inland of the
GBA. It is the emission hotpot of the GBA with a higher $O_3$ concentration (Chen et al., 2022a). HK city is located
at the mouth of the PRD. According to previous source apportionment studies (Li et al., 2012, 2013), the pollution
in HK city is more attributed to the emissions outside the GBA compared to the other cities of the GBA. Regarding
the $O_3$ in the GF region, the Day-0 emission was usually contributed by both local emission and regional transport
within the GBA, with similar contributions. The major source areas of the Day-2 to Day-4 emissions contributing
to $O_3$ in the GF in different episodic cases varied similarly to those contributing to the hourly average $O_3$ in the
GBA. Generally, the influence of local and GBA regional pollutants on $O_3$ in the GF region diminished rapidly
within 1 day. However, the regional emission can still have an important contribution in the episodic case with
southerly winds, such as the $24^{th}$ -$25^{th}$ July (about 26%) in the Ep2 and $23^{rd}$ -$25^{th}$ (about 15%) in the Ep4. For the
$O_3$ in HK city, the local emission amount is low, and its impact was also limited to the current day. In addition,
the $O_3$ in HK city was also susceptible to the impact of pollutants from the ocean but less from the GBA regional
emissions. During the Ep1 periods, it was observed that the contribution of the GBA regional sources largely
increased in the Day-0 emission as the prevailing wind direction shifted to the north. On the other hand,
neighboring provinces' emissions dominated the contributions of emissions from Day-1 to Day-3. Unlike the GF
region, the influence of EC emissions on the $O_3$ in HK was also limited in Ep1. Similar conclusions can be drawn
for the evolution of the spatiotemporal contribution of emissions in Ep3. As discussed above, the $O_3$ pollution in
Ep2 was mainly driven by local emissions. Thus, the $O_3$ concentration in HK city, located in the upwind region
with fewer local emissions, was much lower than the $O_3$ concentration in the GF region. In Ep4, same as the GBA
average and GF region, the impact of pollutants from EC and NCP became important in the Day-2 and Day-3
emissions, which can contribute up to 20% of $O_3$. These results indicate that although $O_3$ is usually considered a
regional pollution problem, it's necessary to consider the local characteristics of different sub-regions when
making more specific prevention and control policies.

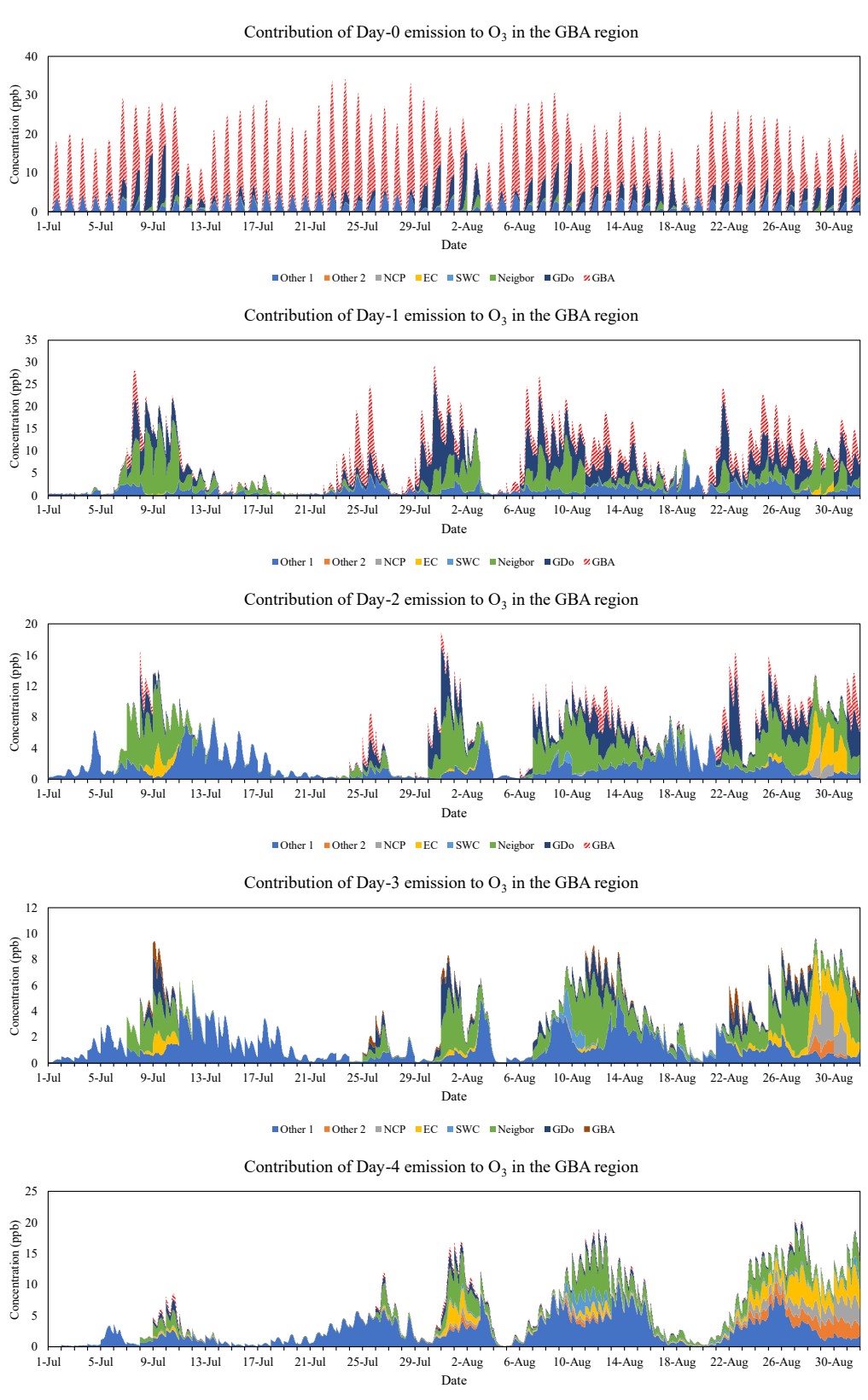


Figure 5. Time series of contributions from different source areas and emitting periods to the O₃ concentrations in the GBA. (*GDo* represents areas outside the GBA region but within Guangdong province. *Neighbor* represents the provinces around Guangdong province. *Other 1* represents ocean and other countries. *Other 2* represents other area within the mainland China in the simulation domain.)

Monthly Average O₃ concentration (ppb)

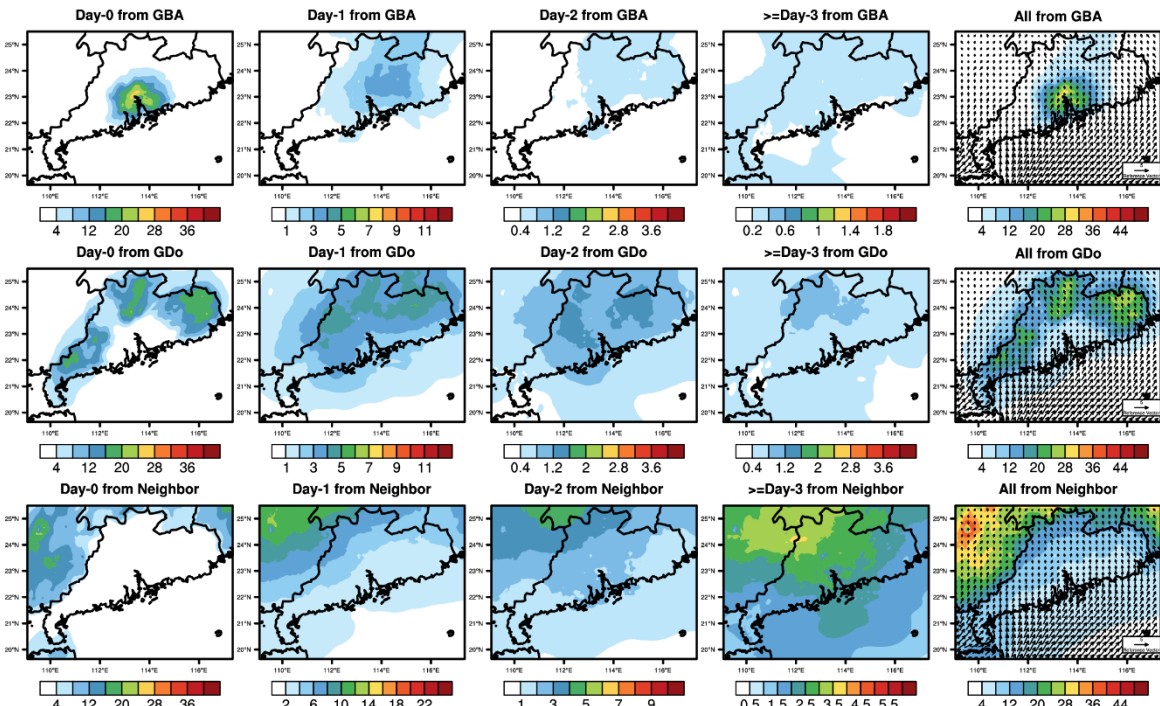

Figure 6. Spatial distribution of monthly average O₃ concentration between 9:00-17:00 (Local time) contributed by emission of GBA, other regions within Guangdong province (GDo), and neighboring provinces (Neighbor) from various periods. (Unit: ppb. Due to the large variation of contribution, the colorbar range of each sub-figure is different)

Average O₃ concentration during 7-10 July (ppb)

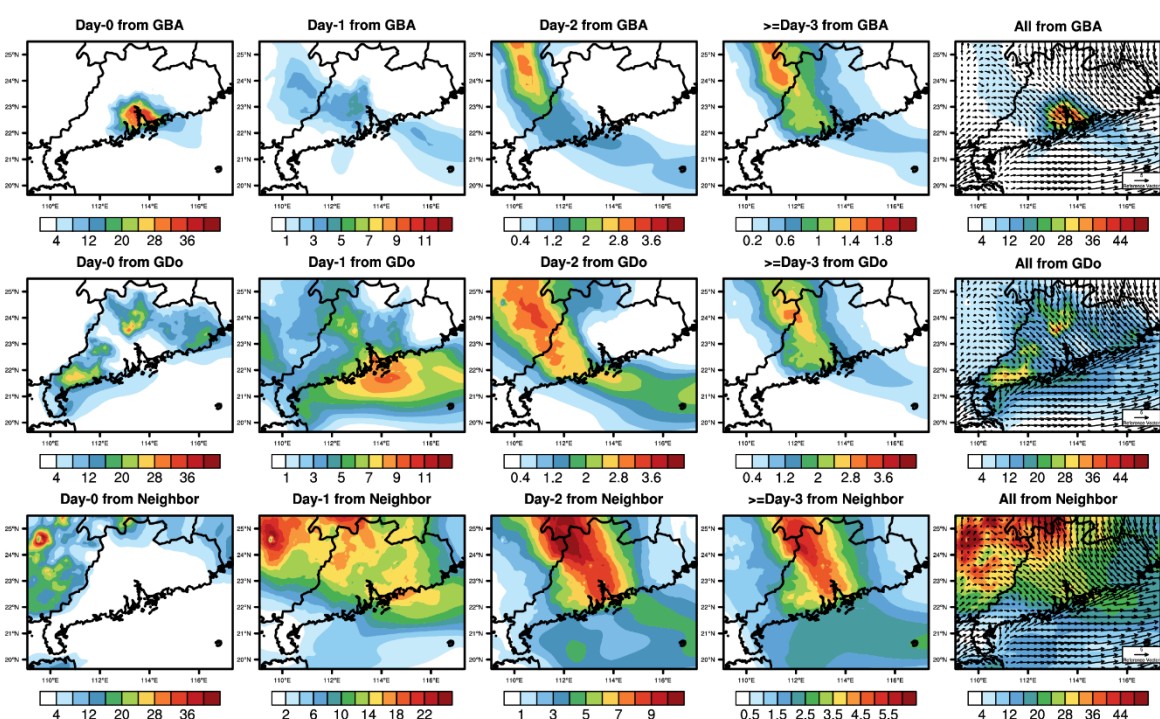

Figure 7. Same as Figure 6, but for the period of 7th-10th July 2016

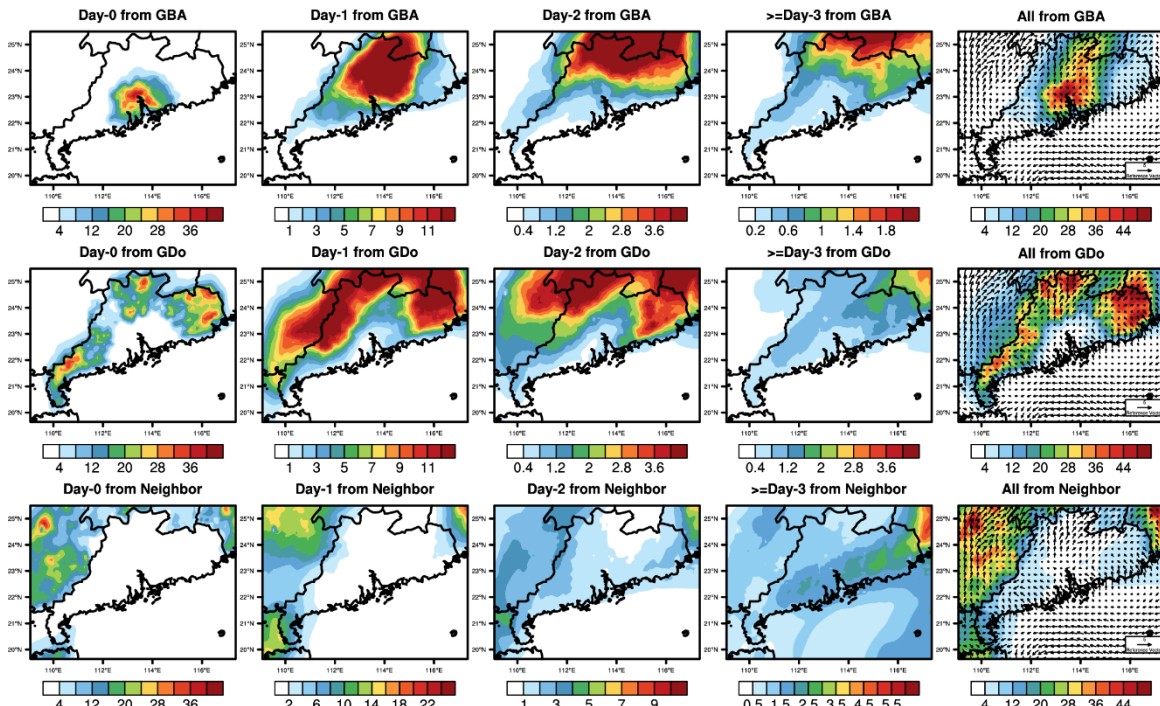

Figure 8. Same as Figure 6, but for the period of 24th-26th July 2016.

### 3.4 Verification of the TSA by comparing to Zero-out Experiments

Here, the emission zero-out sensitivity experiments, another commonly used method for source apportionment, were also conducted to evaluate the results from the TSA method. The zero-out method needs to conduct two sets of simulations, including the control run and the zero-out run. In the control run, the simulations were conducted using the complete emissions. In the zero-out runs, the simulations were conducted with the emissions that specific period and area were removed. Subsequently, the contribution of the specific source area and source time was derived by calculating the difference between the control and zero-out simulations. For each target date, three types of emission area controls were implemented: Type 1 involved the GBA region alone (GBA); Type 2 included the emission within Guangdong province (GD, namely GBA+GDo); and Type 3 expanded to encompass the emission within Guangdong province and neighbouring provinces (GD_Neighbor, namely GBA + GDo + Neighbor). The emission control period was set as continuous control beginning from the current day, which is also the target day (Day-0), from 1 day ago (Day-1), from 2 days ago (Day-2) and from 3 days ago (Day-3), respectively. The zero-out experiments were carried out for the periods between 7th and 10th July (Typhoon case) and between 24th and 26th (Sub-tropical high case). More configurations can be found in Tables S4- S5.

From the result of zero-out experiments (Fig.9 and Fig.S9), it can be seen that, for the typhoon case (Fig.9), when only controlling the emission within the GBA, there is little difference between results of controlling emissions 1 day and 3 days in advance. This is consistent with the TSA result that the influence of the emission within the GBA is usually limited to 2 days. Controlling emissions 1 day in advance in GD yields better results compared to solely controlling emissions within the GBA. There is less variation of the $O_3$ concentration when controlling the emission within GD 2 or 3 days in advance. Meanwhile, regarding only controlling emissions on Day-0, there is limited improvement in controlling the emission for a larger area (GD and GD_Neighbor) than solely within the GBA. This result aligns with the TSA result that the pollutants from neighboring provinces took effect on the $O_3$ over the GBA region at least 1 day later. Joint control from Guangdong and neigboring province has a better optimal effect in the simulations conducted from Day-2 to Day-0 and Day-3 to Day-0. The difference between GD_Neighbor and the GD result is more pronounced in these simulations, indicating that it's more effective to implement joint control within other provinces 2-3 days in advance.

For the sub-tropical high case (Fig.S9), whatever controlling the emissions on the current day or 2 days ahead, the effect of solely controlling emissions within the GBA is similar to those of joint control in a larger area (GD and GD_Neighbor). It supports our previous conclusion that the pollution is mainly contributed by the local sources. Additionally, there is limited optimization effect to control the emission 2-3 days in advance than controlling 1 day in advance. To alleviate this ozone pollution, controlling the local emission in the short term should be effective. Although the contribution discrepancies between the source contribution (%) calculated from the zero-out method and those obtained from the TSA method can reach 20%, which is due to the non-linear chemistry relationship between ozone and its precursors, as well as the differences in the methodology (Kwok et al., 2015; Clappier et al., 2017), similar relationships between source area/time and receptor can be drawn. These results also support the validity of the TSA approach.

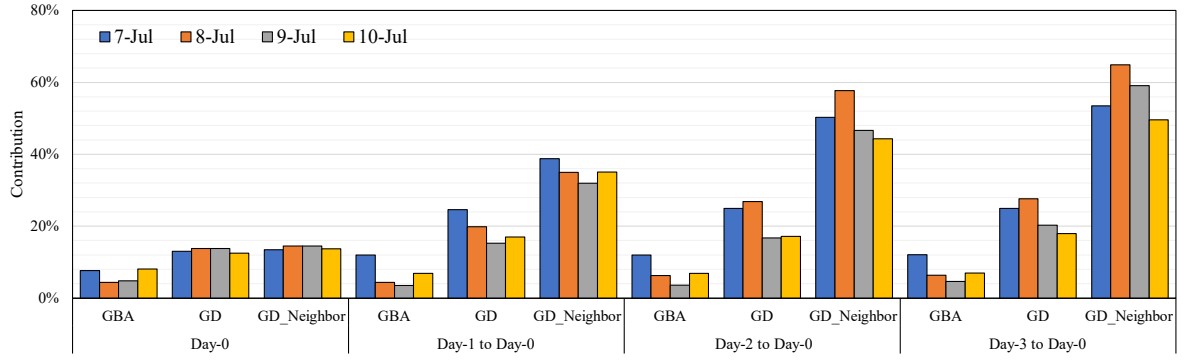

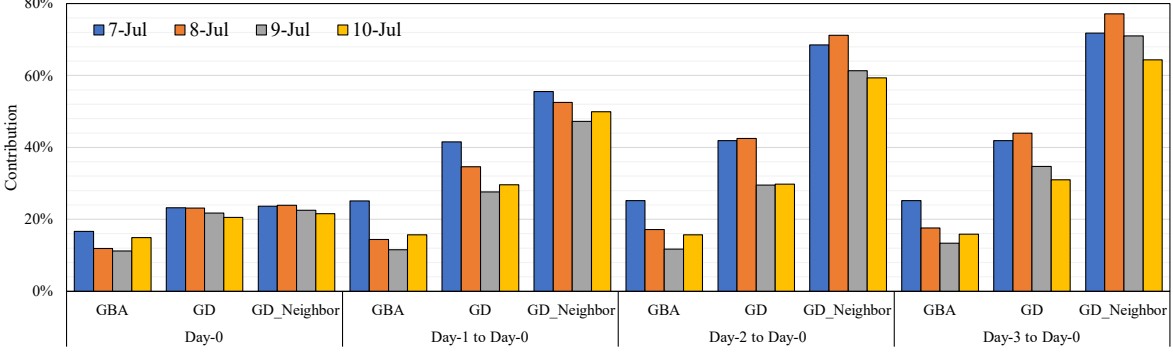

Figure 9. The contribution of different source areas and time periods to the $O_3$ concentration over the GBA in the typhoon case using the zero-out and TSA methods. (Different colors represent different target dates; Upper: Zero-out; Bottom: TSA)

### 3.5 Discussion

Previous studies mainly focused on exploring the contribution and control of various source areas and categories on $O_3$ over the GBA. The analysis in this study illustrated that there could be a larger difference between the temporal contribution of emissions to the $O_3$ pollution over the GBA under different weather patterns. This finding emphasizes the importance of understanding the contribution of pollutants from different emission periods and identifying the major periods, particularly in episodic cases, for effective policymaking in pollution control. In contrast to the zero-out method, which requires multiple simulations, our approach provides a comprehensive overview of source contributions within a single simulation. This method is suited for applications involving more potential sources as it saves computation costs.

In addition, meteorological conditions play an important role in affecting the effectiveness of the emission control area and period. The results here suggest that the approach of typhoons usually strengthens the cross-region transportation of pollutants to the GBA. Therefore, cross-province collaboration and control should be implemented at least 2-3 days ahead when the typhoon is predicted. The information obtained from the TSA results can contribute to the establishment of an early warning and rapid response system. It could help to facilitate collaboration, considering estimated timelines and the cost implications associated with emission reduction efforts, aiming to achieve a balanced outcome across regions. In contrast, local emission control within 2 days is more effective when the GBA is under the influence of a high-pressure system. The primary focus for emission control measures should be on local vehicles and industries, as they are the major contributors of $NO_x$ and VOCs (Bian et al., 2019; Li et al., 2019). Implementing measures such as traffic restrictions based on even- and odd-numbered license plates and temporary reduction of emissions from industries can be effective strategies to target these sources in advance. Our findings emphasize the importance of considering the impact of meteorological conditions when implementing control measures in advance. Here, our study primarily focuses on the summer season, which has been identified as the $O_3$ pollution period in the GBA (Gao et al., 2018; Li et al., 2022). Typhoons and subtropical high-pressure systems are two significant weather patterns closely linked with $O_3$ pollution events in Southern China (Wang et al., 2017; Ouyang et al., 2022). The trajectories of typhoons in episodes 1 and 3 (Figure S3) are similar to one of the typical typhoon pathways, often coinciding with $O_3$ pollution events in the GBA (Qu et al., 2021; Wang et al., 2022a). Meanwhile, the high 2-m temperature and low 2-m relative humidity over the GBA can be observed during the $O_3$ episodes (Figure S10-S11). The prevailing wind across the GBA in the typhoon and sub-tropical high-pressure cases is northerly and southerly, respectively (Figures S12). Overall, the weather conditions observed in the selected cases of this study are similar to those reported in other $O_3$ pollution studies in this region (Qu et al., 2021 Ouyang et al., 2022; Wang et al., 2022). Nevertheless, it is crucial to underscore that the spatial-temporal source contribution may vary in $O_3$ pollutions even under similar meteorological conditions. For instance, the change of typhoon position and intensity could influence the large-scale circulation and precursor emission (Zhan et al., 2020; Wang et al., 2022a). Therefore, it is imperative to undertake further investigations and comparative studies on more similar $O_3$ events over the GBA under the influence of typhoons and subtropical high-pressures in the future, which will contribute to attaining more widely applicable findings and offer valuable insights for developing emission control strategies. Additionally, the spatial-temporal influence of emission to $O_3$ over the GBA under other unfavourable conditions and seasons is also essential to further explore through the TSA method, which helps to gain a more comprehensive understanding of when and where the $O_3$ over the GBA comes from.

In the context of climate change, the occurrence of extreme weather, such as extreme heatwaves (Coffel et al., 2018; Dong et al.,2023), is expected to become more frequent. These events will significantly impact the sources and sinks of pollutants through various physical and chemical processes. At the same time, governments in different countries will implement various emission control strategies in response to climate change, such as carbon neutrality (Liu et al., 2021; Zhang et al., 2021), which will also alter the emission structure. How these extreme weather events and control measures influence the temporal characterization of sources, the formation of air pollution, and the spatial-temporal contribution of emissions from different countries, as well as their interactions, are also worth further investigation in the future. Such investigations can foster mutual cooperation among nations to collectively address environmental challenges.

However, it should be noted that the numerical model source apportionment results are usually influenced by the uncertainties of the emission inventory as most of the emission inventories are constructed by the bottom-up method and cannot be updated in a timely manner. With the increasing availability of different types of observations, including surface monitoring and satellite remote sensing data, different top-down methods such as data assimilation (East et al., 2022) and machine learning (Chen et al., 2023) have been applied to integrate observations and optimize the emissions. These methods should be implemented to update the emission inventory. Meanwhile, the air quality model results are also sensitive to the uncertainty in the weather forecast, potentially leading to variations in source apportionment results. To alleviate the impact of weather forecast uncertainty, different methods, such as ensemble simulation (Gilliam et al., 2015), data assimilation for the meteorological field simulation (Kwon et al., 2018), and machine learning method (Scher et al., 2018; Cho et al., 2020), should be applied to enhance the accuracy of meteorological field simulations.

## 4. Conclusion

In this study, we applied the CAMx-TSA method to analyze the spatial and temporal contribution of different sources to the $O_3$ pollution in the GBA during summer. The result shows that the $O_3$ over the GBA in summer is mainly contributed by the pollutants from local emissions, followed by pollutants originating from other regions within Guangdong province and neighbouring provinces. The $O_3$ formation is predominantly attributed to pollutants emitted within a 3-day period, accounting for over 70% of the total contribution. During the $O_3$ episodes, when the typhoon moved from the eastern Philippine Sea towards southern China, the prevailing wind shifted from south to north over the GBA. This facilitated the transport of pollutants from GDo and neigbouring provinces to the GBA, resulting in an increase in $O_3$ concentrations. The pollutants emitted 3 days ago still have a significant contribution. When the typhoon remained near the sea areas east of Taiwan province and moved northward, under the continuous influence of northerly wind, the emissions from eastern China, even the North China Plain from 3 days ago can also have a noticeable impact on $O_3$ over the GBA. In contrast, when the GBA was mainly under the control of the sub-tropical high-pressure system, the ozone pollution was mainly caused by the local pollutants within the current 2 days. The results indicated that implementing joint emission control measures with other provinces 2-3 days in advance is more effective for preventing the $O_3$ pollution in the GBA when the typhoon is approaching southern China. On the other hand, it's more efficient to pay more attention to local source control within 2 days when the GBA is under the control of the high-pressure system.

Here, different surrounding provinces were categorized as one source area here to save computation resource for more potential source investigation. As the neigbouring province was illustrated as a major contributor to the $O_3$ in the GBA, it is necessary to further divided this source into several sub-source areas and explore their individual impact in future work. Meanwhile, our preliminary findings indicate that pollutants emitted more than three days prior can still have a considerable impact on the $O_3$ levels in the GBA. As a result, it would be valuable to conduct source apportionment analyses with finer source areas and earlier source periods for $O_3$ pollution in different cities within the GBA. This further investigation would provide deeper insights into the unique $O_3$ pollution characteristics of each city. In addition, individual source categories were not separated in this study, mainly due to the application of different emission inventories with different source category classifications, making it difficult to combine them. It is important to note that each source category has its own characteristic temporal profile, which can have different temporal impacts on $O_3$ concentrations. Therefore, the temporal contribution of various source categories, including anthropogenic and biogenic emissions, should be also considered in future work. These works can provide more spatial and temporal information of $O_3$ source over the GBA, enabling local governments to design and implement more targeted control measures more effectively and promptly.

## Code and Data availability

Hourly $O_3$ observation data were released by the China National Environmental Monitoring Centre (http://www.cnemc.cn/en, last access 24 December; CNEMC, 2023) and the Hong Kong Environmental Protection Department (https://cd.epic.epd.gov.hk/EPICDI/air/station/?lang=en, last access 24 December 2023; HKEPD, 2023). The CAMx model code is freely available via https://www.camx.com/download/, last access 24 December, 2023). The ECMWF Reanalysis v5 (ERA5) data was downloaded from https://www.ecmwf.int/en/forecasts/dataset/ecmwf-reanalysis-v5 , last access 17 May 2024; ERA5, 2024)

## Author contribution

CY, LX, and JF designed the research. CY contributed to model development, simulation and data analysis. LX and JF contributed to the result discussion. CY prepared the manuscript with contributions from all co-authors.

## Competing interests

The authors declare that they have no conflict of interest.

**Acknowledgements**

This work was supported by the Research Grants Council of Hong Kong Government (C6026-22GF) and the Improvement on Competitiveness in Hiring New Faculties Funding Scheme of CUHK (No. 4937115)

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
