# Peer review of "Spatiotemporal Source Apportionment of Ozone Pollution over the Greater"

_EGUsphere, 2023_

## Referee Comment (RC2)

In this paper, the authors extended the OSAT of CAMx model to include the function of tracking the lifetime of ozone precursors which then can quantified the temporal contributions of ozone. With the implementation of the method, they analyze the spatial and temporal contributions of difference geographical sources to ozone in GBA in summer in 2016. The research is interesting, but I have some questions which the authors should answer before this paper be published in this journal.

1. In section 2.1, it introduced that the temporal tagging method set a-five-day range. Please show us the reason why you set a-five-range to tracking the air pollutants. Because we can see the contribution of day-4(all the contribution from the days earlier than 3 days ago) to ozone is not small (sometime can reach up to 20%), maybe increase the range can show us more information about the ozone contributions from the four days ago or even earlier.

2. The model outputs are generated by a continuously run simulation. I am curious that what the temporal contribution results will be like if your simulation is segmented run

3. Also, about the temporal tagging method, the chemical production of ozone should both consider the NOx and VOCs. How do you deal with the ozone precursors when they are emitted in different days.

4. Please pay attention to your presentation, the language needs to be more carefully polished.

---

## Author Comment (AC1)

**Reviewer 2**

**The manuscript utilized a set of tools including the Temporal Source Apportionment Method and coupled WRF-CAMx model to quantify the spatial-temporal contribution of precursor emissions under these typical conditions (e.g., typhoons and sub-tropical high-pressure systems) over the Greater Bay Area (GBA) region. The results show that, when a typhoon approaches, $O_3$ pollution is influenced by pollutants emitted 2-3 days earlier from both Guangdong province and its neighboring regions, as well as from eastern China before the 2-day mark. In contrast, local emissions within the 2-day timeframe play a predominant role in contributing to $O_3$ pollution when the GBA is primarily affected by sub-tropical high-pressure systems. Overall, the topic is of interest to the audience and the manuscript is generally well organized. However, before I can only recommend it to be accepted by the EGUsphere journal, the manuscript needs some major revision.**

[Response]: We thank the reviewer for the positive and encouraging comments. In the revised manuscript, we have added more introduction and elaboration of the TSA method, data used, and model setting in the main text and supplemental material. Figures have been replotted with the ERA5 reanalysis data to make them clearer. More discussion of the TSA method application has also been added. Please see our point-to-point response below. The additional content in the revised manuscript is marked with italics in the response. Thank you!

**Major Comment**

**1. Given that many quantitative results are given in this study, the primary concern the author needs to address is whether the results based on three cases are representative of typical weather conditions, especially considering that the results are highly sensitive to study region and episode. According to the research objectives and significance proposed by the authors, aimed at facilitating the development of effective and timely control policies, these quantitative results acquire persuasive and guiding significance only when they achieve statistical significance.**

[Response]: Thanks for your comment and suggestion. In Southern China, typhoons and subtropical high-pressure systems are two significant weather patterns closely linked with $O_3$ pollution events. They have been extensively reported and studied in the region (Wang et al., 2017; Li et al., 2022; Ouyang et al., 2022). The trajectories of typhoons in episodes 1 and 3 of our study (as depicted in Figure R1) are similar to one of the typical typhoon pathways, often coinciding with $O_3$ pollution events in the GBA, as summarized in previous studies (Figure R2, Qu et al., 2021; Wang et al., 2022). As shown in Figure R3, the prevailing wind across the GBA in the typhoon and sub-tropical high-pressure cases at the height of 850 hPa is northerly and southerly, respectively, which are consistent with the results presented by Qu (Figure R4, 2021). As shown in Figure R5-R6, the high 2-m temperature and low 2-m relative humidity over the

GBA can be observed during the $O_3$ episodes, which align with findings of other studies (Ouyang et al., 2022; Wang et al., 2022). Overall, the weather conditions observed in the selected cases of this study are similar to those reported in other $O_3$ pollution studies in this region. Here, our study primarily focuses on summer season, which has been identified as the $O_3$ pollution period in the GBA (Gao et al., 2018; Li et al., 2022). But we completely agree with the reviewer that the TSA results are sensitive to study region and specific episodic periods. There are limitations inherent in the case analysis. Therefore, it is crucial to undertake further investigations and comparative studies on more similar cases, which will contribute to attaining more widely applicable findings and offer valuable insights for developing emission control strategies. We will further explore this issue in future research. We have added more discussion in the revised manuscript. Please find the update in Lines 492-511.

"Our findings emphasize the importance of considering the impact of meteorological conditions when implementing control measures in advance. *Here, our study primarily focuses on the summer season, which has been identified as the $O_3$ pollution period in the GBA (Gao et al., 2018; Li et al., 2022). Typhoons and subtropical high-pressure systems are two significant weather patterns closely linked with $O_3$ pollution events in Southern China (Wang et al., 2017; Ouyang et al., 2022). The trajectories of typhoons in episodes 1 and 3 (Figure S3) are similar to one of the typical typhoon pathways, often coinciding with $O_3$ pollution events in the GBA (Qu et al., 2021; Wang et al., 2022). Meanwhile, the high 2-m temperature and low 2-m relative humidity over the GBA can be observed during the $O_3$ episodes (Figure S10-S11). The prevailing wind across the GBA in the typhoon and sub-tropical high-pressure cases is northerly and southerly, respectively (Figures S12). Overall, the weather conditions observed in the selected cases of this study are similar to those reported in other $O_3$ pollution studies in this region. (Qu et al., 2021 Ouyang et al., 2022; Wang et al., 2022). Nevertheless, it is crucial to underscore that the spatial-temporal source contribution may vary in $O_3$ pollutions even under similar meteorological conditions. For instance, the change of typhoon position and intensity could influence the large-scale circulation and precursor emission (Zhan et al., 2020; Wang et al., 2022). Therefore, it is imperative to undertake further investigations and comparative studies on more similar $O_3$ events over the GBA under the influence of typhoons and subtropical high-pressures in the future, which will contribute to attaining more widely applicable findings and offer valuable insights for developing emission control strategies.* Additionally, the spatial-temporal influence of emission to $O_3$ over the GBA under other unfavourable conditions and seasons is also essential to further explore through the TSA method, which helps to gain a more comprehensive understanding of when and where the $O_3$ over the GBA comes from. "

[Figure]

Figure R1. The moving paths of typhoons during the Ep1, Ep3 and Ep4 $O_3$ episodic cases. (The figures were plotted using the ERA5 reanalysis data.)

[Figure]

Figure R2. (a)Typhoon tracks and its associated abnormally high $O_3$ pollution in city cluster Pear River Delta (Wang et al., 2022). (b) The tracks of typhoons related to $O_3$ pollution in the Pear River Delta in July 2014–2018(Qu et al. 2021).

[Figure]

Figure R3. The wind fields at the height of 850 hPa at 14:00 (Local Time) for O$_3$ episodes in this study (The figures were plotted using the ERA5 reanalysis data).

[Figure]

Figure R4. The wind fields at the height of 850 hPa at 14:00 (Local Time) for O$_3$ episodes (c) summer (typhoon-induced) and (d) summer (no-typhoon). The influenced weather pattern in the no-typhoon cases is subtropical high-pressure system (Table S2 in Qu et al., 2021)

[Figure]

Figure R5. The 2-m temperature at the at 14:00 (Local Time) for O$_3$ episodes in this study. (The figures were plotted using the ERA5 reanalysis data).

[Figure]

Figure R6. The 2-m relative humidity at the at 14:00 (Local Time) for O₃ episodes in this study. (The figures were plotted using the ERA5 reanalysis data).

Reference

Gao, X., Deng, X., Tan, H., Wang, C., Wang, N., & Yue, D. (2018). Characteristics and analysis on regional pollution process and circulation weather types over Guangdong Province. *Acta Scientiae Circumstantiae*, *38*(5), 1708-1716. (in Chinese)

Li, T. Y., Chen, J. Y., Weng, J. F., Shen, J., & Gong, Y. (2022). Ozone pollution synoptic patterns and their variation characteristics in Guangdong Province. *China Environmental Science*, *42*(5), 2015-2024. (in Chinese)

Li, Y., Zhao, X., Deng, X., & Gao, J. (2022). The impact of peripheral circulation characteristics of typhoon on sustained ozone episodes over the Pearl River Delta region, China. *Atmospheric Chemistry and Physics*, *22*(6), 3861-3873.

Ouyang, S., Deng, T., Liu, R., Chen, J., He, G., Leung, J. C. H., ... & Liu, S. C. (2022). Impact of a subtropical high and a typhoon on a severe ozone pollution episode in the Pearl River Delta, China. *Atmospheric Chemistry and Physics*, *22*(16), 10751-10767.

Qu, K., Wang, X., Yan, Y., Shen, J., Xiao, T., Dong, H., ... & Zhang, Y. (2021). A comparative study to reveal the influence of typhoons on the transport, production and accumulation of $O_3$ in the Pearl River Delta, China. *Atmospheric Chemistry and Physics*, *21*(15), 11593-11612.

Wang, N., Huang, X., Xu, J., Wang, T., Tan, Z. M., & Ding, A. (2022). Typhoon-boosted biogenic emission aggravates cross-regional ozone pollution in China. *Science Advances*, *8*(2), eabl6166.

Wang, T., Xue, L., Brimblecombe, P., Lam, Y. F., Li, L., & Zhang, L. (2017). Ozone pollution in China: A review of concentrations, meteorological influences, chemical precursors, and effects. *Science of the Total Environment*, *575*, 1582-1596.

Zhan, C., Xie, M., Huang, C., Liu, J., Wang, T., Xu, M., ... & Nie, D. (2020). Ozone affected by a succession of four landfall typhoons in the Yangtze River Delta, China: major processes and health impacts. *Atmos. Chem. Phys*, *20*, 13781-13799.

**2. Please elaborate on the criteria utilized in defining sub-regions within Guangdong Province.**

[Response]: Thanks for your comment. The division of sub-regions within the GBA region is primarily based on administrative boundaries and their geographical locations. The sub-regions mainly consist of neighboring cities. Guangzhou and Foshan (GF) are located at the central of the GBA, Shenzhen and Dongguan (SD) are located at the eastern of the GBA, Zhongshan, Zhuhai, and Jiangmen (ZZJ) are located at the southwestern part of the GBA. Zhaoqing and Huizhou (GBAo) have the lowest emission density and are located at the northwestern and northeastern corners, respectively. According to previous source apportionment studies (Li et al., 2012; Chen et al., 2022a; Chen et al., 2022b), the air pollutants in Hong Kong were usually more influenced by long-range transport from regions outside the GBA, in contrast to the other cities in the GBA. Hence, Hong Kong city (HK) is treated as a separate entity.

In this study, our primary objective is to conduct a preliminarily investigation into the overall spatial and temporal sources contribution to the $O_3$ concentration in the GBA region. But we acknowledge the importance of conducting $O_3$ source apportionment for individual cities as well. The results would enable the implementation of more targeted control measures tailored to the specific characteristics of each city. It is worth further investigating in future work. We have added additional explanation in the model setting and discussion. Please refer to Line 166-172 and Line 554-558 in the revised manuscript.

"*The cities within the GBA were separated into different sub-regions mainly based on administrated boundaries and their geographical location, same as the work of Chen et al. (2022c). The sub-regions mainly consist of neighboring cities. Zhaoqing and Huizhou, located at the northwestern and northeastern corners, respectively, were categorized into one group since they have a relatively lower emission density than other cities. Previous studies indicated that the air pollutants in Hong Kong were usually more influenced by long-range transport*

*from regions outside the GBA, in contrast to the other cities in the GBA (Li et al., 2012; Chen et al., 2022a; Chen et al., 2022c). Hence, Hong Kong city (HK) is treated as a separate entity."*

*"Meanwhile, our preliminary findings indicate that pollutants emitted more than three days prior can still have a considerable impact on the $O_3$ levels in the GBA. As a result, it would be valuable to conduct source apportionment analyses with finer source areas and earlier source periods for $O_3$ pollution in different cities within the GBA. This further investigation would provide deeper insights into the unique $O_3$ pollution characteristics of each city."*

Reference

Chen, W., Chen, Y., Chu, Y., Zhang, J., Xi, C., Lin, C., ... & Lu, X. (2022a). Numerical Simulation of Ozone Source Characteristics in the Pearl River Delta Region. *Acta Scientiae Circumstantiae, 42*(3), 293-308. (in Chinese)

Chen, Y., Fung, J. C., Huang, Y., Lu, X., Wang, Z., Louie, P. K., ... & Lau, A. K. (2022b). Temporal source apportionment of $PM_{2.5}$ over the Pearl River Delta region in southern China. *Journal of Geophysical Research: Atmospheres*, *127*(14), e2021JD035271.

Li, Y., Lau, A. H., Fung, J. H., Zheng, J. Y., Zhong, L. J., & Louie, P. K. K. (2012). Ozone source apportionment (OSAT) to differentiate local regional and super-regional source contributions in the Pearl River Delta region, China. *Journal of Geophysical Research: Atmospheres*, *117*(D15).

**3. Please introduce the input data used in the TSA method including details such as sources, resolution, etc. Furthermore, please provide an overview of the fundamental workflow for the TSA method. Specifically, are 'Precursor Tracer Day-x' and 'O₃ Tracer Day-x' calculated simultaneously or sequentially? It would be preferable to use numbers in Fig. 1 to indicate the logical sequence.**

**[Response]:** Thanks for your comment. The input data used in the TSA method include the source area map and hourly emission data. The source area map assigns each model grid cell to one of the specific source regions as defined in Figure 2. The hourly emission data is same as the one used in the normal simulation without source apportionment. For the region outside the GBA, the anthropogenic emission data with a resolution of 0.25° is developed by the MEIC group in the Tsinghua University. For the region within the GBA, the anthropogenic emission data is provided by the HKEPD and the resolution is 3km. The biogenic emission for the entire simulation domain was generated by the MEGAN model.

The TSA method was developed based on the OSAT method. Hence, its workflow is the same as that of the OSAT method. In the TSA method, the tracers go through all processes, including emission, transport, diffusion, and chemical reactions sequentially in each time step, similar to the normal model simulation. Therefore, the *Precursors and $O_3$ tracers* that tracked different periods are calculated simultaneously. When the pollutants emitted from the sources, they will be assigned to the *Precursors Tracers* in Day-0, while the *Precursors Tracers* that tracked other periods and the *$O_3$ Tracers* remain unchanged. The data transfer between tracers (e.g., *Day-1*

to *Day-2*, and *Day-0* to *Day-1*, dash arrow in Figure 1) will be conducted once after one day's simulation. We have added more introduction of the TSA method and data used in the manuscript. Please refer to section 2.1 in the revised manuscript.

"Previously, we have successfully implemented the $PM_{2.5}$ temporal source apportionment method in the CAMx model and applied it to investigate the temporal influence of emissions on $PM_{2.5}$ in the GBA (Chen et al., 2022c). Here, we further extend this method to track the temporal contribution of emissions to the precursors and the formation of $O_3$. *Similar to the OSAT method, the input data used in the TSA method developed in this work include the source area map and hourly emission data. The source area map assigns each model grid cell to one of the specific source regions. The hourly emission data is same as the one used in the normal CAMx model simulation without turning on the source apportionment module.* The basic mechanism of the TSA method is to track the contribution of pollutants from different emitting periods using a set of tracers. In the TSA method (Fig. 1), the *Precursor Tracer Day-x* was used to track the precursors emitted from *x* days ago. The *$O_3$ Tracer Day-x* was used to track the $O_3$ formed from the precursors emitted from corresponding *x* days ago (namely *Precursor Tracer Day-x*). The tracers in *Day-x* can be set into different finer periods (e.g., every 1 hour, 6 hours, 24 hours) as required. The total number of tracers will be decided according to the entire tracking period and the minimum tracking period per tracer. For instance, if the entire tracking period is 5 days and the minimum tracking period per tracer is every 6 hours, the total number of tracers will be 20. *In each time step, the tracers go through all the processes, including emission, transport, diffusion, and chemical reactions sequentially as in the normal CAMx model simulation. Therefore, the Precursors and $O_3$ tracers that tracked different periods are calculated simultaneously. When the pollutants emitted from the sources, they will be assigned to the Precursors Tracer in Day-0, while the Precursors Tracers that tracked other periods and the $O_3$ tracers remain unchanged. The data transfer between tracers (e.g., Day-1 to Day-2, and Day-0 to Day-1, dash arrow in Figure 1) will be conducted once after one day's simulation.* As shown in the Figure 1, during each day's simulation, the contribution of present day's emission is consistently tracked by the Day-0 tracers. After completing the current day's simulation and before starting the next day's simulation, each tracer *Day-x*'s value transfers to the corresponding tracer *Day-(x+1)*, which represents one day earlier than *Day-x*, following the specified sequence. For example, beginning from the penultimate tracer, namely values in Day-3 transfer and add into Day-4, then the values in Day-2 transfer to Day-3, followed by Day-1 to Day-2, and lastly Day-0 to Day-1 (Dash arrow in Figure 1). Here, the value in Day-3 tracer will be added into the last tracer (Day-4) because the last tracer represents the total contribution of pollutants emitted earlier than 3 days ago. "

**4. In the discussion section, the authors suggested that the findings can provide more spatial and temporal information of O₃ sources over the GBA, enabling local governments to design and implement targeted control measures more effectively and promptly. Further discussion is needed for more specific control strategies and policies, along with addressing the underlying potential problems and difficulties, such as the impact of the uncertainty in weather forecasts on final results, and obstacles in cross-regional governance.**

[Response]: Thanks for your comment and suggestions. We have added more discussion for the specific emission control policies and uncertainty of weather forecast in the discussion. Please refer to Line 484-492 and Line 526-531 in the revised manuscript.

"In addition, meteorological conditions play an important role in affecting the effectiveness of the emission control area and period. The results here suggest that the approach of typhoons usually strengthens the cross-region transportation of pollutants to the GBA. Therefore, cross-province collaboration and control should be implemented at least 2-3 days ahead when the typhoon is predicted. *The information obtained from the TSA results can contribute to the establishment of an early warning and rapid response system. It could help to facilitate collaboration, considering estimated timelines and the cost implications associated with emission reduction efforts, aiming to achieve a balanced outcome across regions.* In contrast, local emission control within 2 days is more effective when the GBA is under the influence of a high-pressure system. *The primary focus for emission control measures should be on local vehicles and industries, as they are the major contributors of NOx and VOCs (Bian et al., 2019; Li et al., 2019). Implementing measures such as traffic restrictions based on even- and odd-numbered license plates and temporary reduction of emissions from industries can be effective strategies to target these sources in advance.*"

"However, it should be noted that the numerical model source apportionment results are usually influenced by the uncertainties of the emission inventory as most of the emission inventories are constructed by the bottom-up method and cannot be updated in a timely manner. With the increasing availability of different types of observations, including surface monitoring and satellite remote sensing data, different top-down methods such as data assimilation (East et al., 2022) and machine learning (Chen et al., 2023) have been applied to integrate observations and optimize the emission. These methods should be implemented to update the emission inventory. *Meanwhile, the air quality model results are also sensitive to the uncertainty in the weather forecast, potentially leading to variations in source apportionment results. To alleviate the impact of weather forecast uncertainty, different methods, such as ensemble simulation (Gilliam et al., 2015), data assimilation for the meteorological field simulation (Kwon et al., 2018), and machine learning method (Scher et al., 2018; Cho et al., 2020), should be applied to enhance the accuracy of meteorological field simulations.*"

Reference

Bian, Y., Huang, Z., Ou, J., Zhong, Z., Xu, Y., Zhang, Z., ... & Zheng, J. (2019). Evolution of anthropogenic air pollutant emissions in Guangdong Province, China, from 2006 to 2015. *Atmospheric Chemistry and Physics*, *19*(18), 11701-11719.

Cho, D., Yoo, C., Im, J., & Cha, D. H. (2020). Comparative assessment of various machine learning-based bias correction methods for numerical weather prediction model forecasts of extreme air temperatures in urban areas. *Earth and Space Science*, *7*(4), e2019EA000740.

Gilliam, R. C., Hogrefe, C., Godowitch, J. M., Napelenok, S., Mathur, R., & Rao, S. T. (2015). Impact of inherent meteorology uncertainty on air quality model predictions. *Journal of Geophysical Research: Atmospheres*, *120*(23), 12-259.

Kwon, I. H., English, S., Bell, W., Potthast, R., Collard, A., & Ruston, B. (2018). Assessment of progress and status of data assimilation in numerical weather prediction. *Bulletin of the American Meteorological Society*, *99*(5), ES75-ES79.

Li, M., Zhang, Q., Zheng, B., Tong, D., Lei, Y., Liu, F., ... & He, K. (2019). Persistent growth of anthropogenic non-methane volatile organic compound (NMVOC) emissions in China during 1990–2017: drivers, speciation and ozone formation potential. *Atmospheric Chemistry and Physics*, *19*(13), 8897-8913.

Scher, S., & Messori, G. (2018). Predicting weather forecast uncertainty with machine learning. *Quarterly Journal of the Royal Meteorological Society*, *144*(717), 2830-2841.

**Specific comments:**

**1. Line 111: "TSA" has already been defined in Line 105.**

**[Response]:** Thanks for your comment. We have removed "TSA" in this sentence.

**2. Line 113: "Here, we further extend this method to track the temporal contribution of emissions to $O_3$ and its precursors." Please clarify the sentence for accurate expression given that $O_3$ is not directly emitted from sources as a secondary air pollutant.**

**[Response]:** Thanks for your comment. We have rewritten this sentence to make the expression accurately. Please refer to Line 114 in the revised manuscript.

"*Here, we further extend this method to track the temporal contribution of emissions to the precursors and the formation of $O_3$.*"

**3. Line 127: "add into" -> "be added into"**

**[Response]:** Thanks for your comment. We have corrected it.

**4. Line 159: "Day3" -> "Day-3"**

**[Response]:** Thanks for your comment. We have corrected it.

**5. Line 160: "Day-4 the total..."-> "Day-4 represents the total..."**

**[Response]:** Thanks for your comment. It has been corrected.

**6. Line 167: The temporal resolution of the variables should be stated (hourly or daily).**

**[Response]:** Thanks for your comment. The temporal resolution of the variables is hourly. We have stated it in the manuscript. Please refer to Line 185 in the revised manuscript.

"*The performance of simulated hourly 2-m temperature, 10-m wind speed, and $O_3$ concentration were evaluated and shown in Table S1.* "

**7. Please provide the definition and corresponding mathematical formula for evaluation metrics (e.g., index of agreement) in section 2.2.**

**[Response]:** Thanks for your comment and suggestion. The definition and corresponding mathematical formula of the evaluation metrics were added. Please refer to Line 188 in the revised manuscript and Table S6 in the supplemental material.

"*Here, the statistical metrics, including mean bias (MB), normalized mean bias (NMB), index of agreement (IOA) and root mean square error (RMSE), were used for model performance evaluation. The mathematical formulas for these metrics can be found in Table S6.* "

Table S6 The mathematical formula of statical metrices.

| Statistic Metric | Formular |
|------------------|----------|
| Mean bias (MB) | $\frac{1}{n}\sum_{i=1}^{n}(Mod_i - Obs_i)$ |
| Normalized mean bias (NMB) | $\frac{\sum_{i=1}^{n}(Mod_i - Obs_i)}{\sum_{i=1}^{n} Obs_i}$ |
| Index of agreement (IOA) | $1 - \frac{\sum_{i=1}^{n}(Mod_i - Obs_i)^2}{\sum_{i=1}^{n}(|Mod_i - \overline{Obs}| + |Obs_i - \overline{Obs}|)^2}$ |
| Root mean square error (RMSE) | $\sqrt{\frac{1}{n}\sum_{i=1}^{n}(Mod_i - Obs_i)^2}$ |

*"Here, n is the total numbers of observations. Obs is the observation. Mod is the model result. $\overline{Obs}$ is the average of observations. The MB, NMB, and RMSE are applied to evaluate how well models capture the magnitude of observations. The IOA is applied to evaluate how well models capture the variations in observations (Huang et al., 2021). "*

Reference:

Huang, L., Zhu, Y., Zhai, H., Xue, S., Zhu, T., Shao, Y., ... & Li, L. (2021). Recommendations on benchmarks for numerical air quality model applications in China–Part 1: $PM_{2.5}$ and chemical species. *Atmospheric Chemistry and Physics*, *21*(4), 2725-2743.

**8. Line 181: "There were several O₃ episodes occurred during the simulation period." ->"There were several O₃ episodes that occurred during the simulation period."**

**[Response]:** Thanks for your comment. We have rewritten this sentence.

*"There were several $O_3$ episodes that occurred during the simulation period. "*

**9. Line 181: "the 8-h maximum O₃ concentration (MDA8)" -> "the maximum daily 8-h average (MDA8) O₃ concentration".**

**[Response]:** Thanks for your comment. It has been revised.

**10. Please elaborate on the standards for identifying ozone episodes.**

**[Response]:** Thanks for your comment. The ozone episode was identified based on the average maximum daily 8-h average (MDA8) $O_3$ observation concentrations over the whole GBA. A day was considered a pollution day when the average MDA8 $O_3$ concentrations over the GBA exceeded the standard of 80 ppb (Wang et al., 2022). To capture the evolution of the $O_3$ pollution, based on the characteristics of concentration variation, the days preceding and following the $O_3$ pollution day have also been included for the analysis. For instance, for the $O_3$ episode between $7^{th}$ and $10^{th}$ July, the MDA8 $O_3$ concentration surpassed 80 ppb during $8^{th}$-$9^{th}$ July. In order to better analyze the variations in source characteristics throughout the entire pollution period, $7^{th}$ and $10^{th}$ July have also been included in the analysis. We have added the identification standard of ozone episode in the manuscript. Please refer to Line 204-207 in the revised manuscript.

*"Here, pollution days are identified when the average MDA8 $O_3$ observation concentrations over the GBA exceeded 80ppb (Wang et al., 2022). To better capture the evolution of the $O_3$ pollution, based on the characteristics of concentration variation, the days preceding and following the $O_3$ pollution days were also included in the analysis and the whole period was considered as an $O_3$ episode."*

Reference

Wang, W., Parrish, D. D., Wang, S., Bao, F., Ni, R., Li, X., ... & Su, H. (2022). Long-term trend of ozone pollution in China during 2014–2020: distinct seasonal and spatial characteristics and ozone sensitivity. *Atmospheric chemistry and physics*, *22*(13), 8935-8949.

**11. For clarity, please use a single dashed box to represent one O₃ episode, while representing typhoons and high-pressure events with two distinctive colors in Fig. 3.**

[**Response**]: Thanks for your suggestion. We have replotted Figure 3 to enhance clarity. The typhoon and sub-tropical high-pressure events were represented by boxes with blue and yellow colors, respectively.

[Figure]

Figure 3. The time-series of the observed and simulated MDA8 O₃ concentration over the GBA during July-August 2016 and the synoptic patterns during the O₃ episodes. (Blue box: typhoon case; Yellow box: sub-tropical high-pressure case. The O₃ observations were obtained from the CNEMC and the HKEPD. The synoptic patterns were plotted using the ERA5 reanalysis data)

**12. Please plot the figures (Fig. 3 & Fig. S3) with the reanalysis data (e.g. ERA5) with a satisfactory resolution, Additionally, clarify the source of O₃ in the caption of Fig. 3 and include the results of model comparisons.**

[**Response**]: Thanks for your suggestion. The figures have been replotted with the ERA5 reanalysis data. The source of O₃ observation data and model results have been added. Please refer to Line 203 and Figure 3 in the revised manuscript and Figure S3 in the supplemental material.

*"The O₃ observations were obtained from the China National Environmental Monitoring Centre (CNEMC) and the HKEPD. "*

[Figure]

Figure 3. The time-series of the observed and simulated MDA8 O₃ concentration over the GBA during July-August 2016 and the synoptic patterns during the O₃ episodes. (Blue box: typhoon case; Yellow box: sub-tropical high-pressure case. The O₃ observations were obtained from the CNEMC and the HKEPD. The synoptic patterns were plotted using the ERA5 reanalysis data)

[Figure]

Figure S3. The moving paths of typhoons during the Ep1, Ep3 and Ep4 O₃ episodic cases. (The figures were plotted using the ERA5 reanalysis data. The data was download from https://www.ecmwf.int/en/forecasts/dataset/ecmwf-reanalysis-v5 )

**13. Line 205: Please use "average MDA8 O₃" instead of "average O₃" if MDA8 O₃ concentrations are used and maintain consistency throughout the entire text.**

[Response]: Thanks for your comment. The data used here is the average of the hourly O₃ concentration. We have revised it to "average hourly O₃ concentration" to avoid confusion.

**14. Line 233: "because with" -> "because of".**

[Response]: Thanks for your comment. We have corrected it.

**15. Line 236: Please revise this sentence to ensure its semantic accuracy.**

[Response]: Thanks for your comment. We have rewritten the sentence to ensure accurate expression.

*"The contribution of emission from the GDo and neighboring provinces to $O_3$ concentration in GF, SD, ZZJ regions, and HK city increased by 27%, 21%, 32%, and 22%, respectively. "*

**16. Line 245: "accounting for" -> "accounted for".**

[Response]: Thanks for your comment. We have corrected it.

**17. Line 414: "...control area was set as the GBA, Guangdong province (GD), Guangdong..." Guangdong province has already been defined as GDo in Line 208. Please do not redefine variables and maintain consistency in their definitions. The same problem should be checked throughout the entire text.**

[Response]: Thanks for your comment and suggestion. We are sorry for the confusion caused. As the GBA is part of Guangdong province, we define the other regions within the Guangdong province apart from the GBA as "GDo". Therefore, in the zero-out experiments, we set different types of source area control: Type 1 controls emission within the GBA only (GBA); Type 2 controls the emission within Guangdong province (GD, namely GBA+GDo); Type 3 controls the emission from Guangdong and neighbouring provinces (GD_Neighbor, namely GBA + GDo + Neighbor). We have added more explanation in the revised manuscript. Please refer to Line 163-166 and Line 436-439.

*"Other region within Guangdong province but outside the GBA(GDo), different subregions within the GBA: Guangzhou and Foshan(GF), Shenzhen and Dongguan(SD), Hong Kong (HK), Zhuhai, Zhongshan and Jiangmen (ZZJ), Zhaoqing and Huizhou(GBAo). "*

*" For each target date, three types of emission area controls were implemented: Type 1 involved the GBA region alone (GBA); Type 2 included the emission within Guangdong province (GD, namely GBA+GDo); and Type 3 expanded to encompass the emission within Guangdong province and neighbouring provinces (GD_Neighbor, namely GBA + GDo + Neighbor). "*

**18. Line 434: "GD_neigh" should be used after it has been defined. The same problem should be checked throughout the entire text.**

[**Response**]: Thanks for your comment and suggestion. We are also sorry for the mistake here. We have added the definitions of "GD_Neighbor" in Line 436-439 in the revised manuscript. We have thoroughly reviewed the manuscript and made corrections for similar issues.

" *For each target date, three types of emission area controls were implemented: Type 1 involved the GBA region alone (GBA); Type 2 included the emission within Guangdong province (GD, namely GBA+GDo); and Type 3 expanded to encompass the emission within Guangdong province and neighbouring provinces (GD_Neighbor, namely GBA + GDo + Neighbor).* "

**19. If the same abbreviation is defined in the caption of one figure (in the first occurrence in the figure caption), there is no need for redefinition in subsequent figures (e.g., Fig. 6 & 7 & 8). The problem should be checked throughout the entire text.**

[**Response**]: Thanks for your suggestion. We have revised the captions of figures in the whole manuscript to avoid duplicating definitions.

*Figure 6. Spatial distribution of monthly average $O_3$ concentration between 9:00-17:00 (Local time) contributed by emission of GBA, other regions within Guangdong province (GDo), and neighboring provinces (Neighbor) from various periods. (Unit: ppb. Due to the large variation of contribution, the colorbar range of each sub-figure is different)*

*Figure 7. Same as Figure 6, but for the period of $7^{th}$-$10^{th}$ July 2016.*

*Figure 8. Same as Figure 6, but for the period of $24^{th}$-$26^{th}$ July 2016.*

**20. The language should be carefully refined before it is accepted.**

[**Response**]: Thanks for your comment and suggestion. We have thoroughly reviewed and checked the English writing of the manuscript. For example, in the discussion section,

" *Previous studies mainly focused on exploring the contribution and control of various source areas and categories on $O_3$ over the GBA. The analysis in this study illustrated that there could be a larger difference between the temporal contribution of emissions to the $O_3$ pollution over the GBA under different weather patterns. This finding emphasizes the importance of understanding the contribution of pollutants from different emission periods and identifying the major periods, particularly in episodic cases, for effective policymaking in pollution control. In contrast to the zero-out method, which requires multiple simulations, our approach provides a comprehensive overview of source contributions within a single simulation. This method is suited for applications involving more potential sources as it saves computation costs.* "

"*In the context of climate change, the occurrence of extreme weather, such as extreme heatwaves (Coffel et al., 2018; Dong et al.,2023), is expected to become more frequent. These events will significantly impact the sources and sinks of pollutants through various physical and chemical processes. At the same time, governments in different countries will implement various emission control strategies in response to climate change, such as carbon neutrality (Liu et al., 2021; Zhang et al., 2021), which will also alter the emission structure. How these extreme weather events and control measures influence the temporal characterization of sources, the formation of air pollution, and the spatial-temporal contribution of emissions from different countries, as well as their interactions, are also worth further investigation in the future. Such investigations can foster mutual cooperation among nations to collectively address environmental challenges.* "

---

## Author Comment (AC3)

**Reviewer 1**

**In this paper, the authors extended the OSAT of CAMx model to include the function of tracking the lifetime of ozone precursors which then can quantified the temporal contributions of ozone. With the implementation of the method, they analyze the spatial and temporal contributions of difference geographical sources to ozone in GBA in summer in 2016. The research is interesting, but I have some questions which the authors should answer before this paper be published in this journal.**

**[Response]:** We thank the reviewer for the positive and encouraging comments. In the revised manuscript, we have added more elaboration of the TSA method and model setting in the main text. More discussion of the TSA method application has also been added. Please see our point-by-point response below. The additional content in the revised manuscript is marked with italics in the response. Thank you!

**1. In section 2.1, it introduced that the temporal tagging method set a-five-day range. Please show us the reason why you set a-five-range to tracking the air pollutants. Because we can see the contribution of day-4(all the contribution from the days earlier than 3 days ago) to ozone is not small (sometime can reach up to 20%), maybe increase the range can show us more information about the ozone contributions from the four days ago or even earlier.**

**[Response]:** Thanks for your comment. The primary objective of this study is to examine the collective spatial-temporal impact of different sources on $O_3$ levels in the GBA region. By utilizing the TSA method, the tracers employed will be N times greater than those used in the OSAT method when set to a-N-range. In this study, we have chosen to set it to a-five-range to assess the influence of a broader range of source areas, taking into account computational resources. In addition, Figure 5 shows the time series of contributions from different source areas and emitting periods to the $O_3$ concentrations in the GBA. It shows that the influence of pollutants emitted earlier than 3 days ago was much smaller in July. Usually, the pollutants emitted earlier than 3 days ago originate from the ocean and other countries. However, we agree with the reviewer that it's necessary to investigate further the contribution of pollutants emitted from earlier periods, which will provide more information, such as $O_3$ concentration in late August. The preliminary result of this study could provide a reference for a more comprehensive and targeted investigation in future work. We have clarified it in the discussion section. Please refer to Line 554-558 in the revised manuscript.

*"Meanwhile, our preliminary findings indicate that pollutants emitted more than three days prior can still have a considerable impact on the $O_3$ levels in the GBA. As a result, it would be valuable to conduct source apportionment analyses with finer source areas and earlier source periods for $O_3$ pollution in different cities within the GBA. This further investigation would provide deeper insights into the unique $O_3$ pollution characteristics of each city."*

**2. The model outputs are generated by a continuously run simulation. I am curious that what the temporal contribution results will be like if your simulation is segmented run.**

[Response]: Thanks for your comment. The model results were generated by the segmented simulation. We conducted one day simulation each time. The simulation result of the last hour in the current day's simulation will be used as the initial conditions for the next day's simulation. Hence, in theory, the model results obtained from the segmented run are identical to those obtained from the continuous run.

**3. Also, about the temporal tagging method, the chemical production of ozone should both consider the NOx and VOCs. How do you deal with the ozone precursors when they are emitted in different days.**

[Response]: Thanks for your comment. In the OSAT method, the sensitivity of $O_3$ formation is determined by the photochemical indicator, the ratio of the production rate of hydrogen peroxide ($H_2O_2$) and nitric acid ($HNO_3$) (denoted as $P(H_2O_2)/P(HNO_3)$). When the $P(H_2O_2)/P(HNO_3) > 0.35$, the $O_3$ formation is classified as NOx-limited. In this case, the contributions to ozone formation are attributed to different NOx sources based on the proportion of their emissions to the total NOx emissions. Conversely, when the $O_3$ formation is classified as VOC-limited, the contributions are proportionally attributed to the sources of VOCs. Hence, the TSA method follows the same rule to determine the sensitivity of $O_3$ formation. When it is NOx-limited, the contributions are proportionally distributed to the NOx sources emitted on specific days. Same method is applied to VOCs sources when it is VOC-limited. We have added more description of the $O_3$ formation sensitivity in the TSA method in Line 138-142 in the revised manuscript.

"*Same as the OSAT method, the TSA method also utilizes the photochemical indicator, namely, the ratio of the production rate of hydrogen peroxide ($H_2O_2$) and nitric acid ($HNO_3$), to determine the sensitivity of $O_3$ formation. When the $O_3$ formation is classified as NOx-limited (VOC-limited), the contributions are distributed to the NOx (VOCs) sources emitted at different periods, based on the proportion of their emissions to the total NOx (VOCs) emissions.*"

**4. Please pay attention to your presentation, the language needs to be more carefully polished.**

[Response]: Thanks for your comment and suggestion. We have thoroughly reviewed and checked the English writing of the manuscript. For example, in the discussion section,

" *Previous studies mainly focused on exploring the contribution and control of various source areas and categories on $O_3$ over the GBA. The analysis in this study illustrated that there could be a larger difference between the temporal contribution of emissions to the $O_3$ pollution over the GBA under different weather patterns. This finding emphasizes the importance of*

*understanding the contribution of pollutants from different emission periods and identifying the major periods, particularly in episodic cases, for effective policymaking in pollution control. In contrast to the zero-out method, which requires multiple simulations, our approach provides a comprehensive overview of source contributions within a single simulation. This method is suited for applications involving more potential sources as it saves computation costs.* "

"*In the context of climate change, the occurrence of extreme weather, such as extreme heatwaves (Coffel et al., 2018; Dong et al.,2023), is expected to become more frequent. These events will significantly impact the sources and sinks of pollutants through various physical and chemical processes. At the same time, governments in different countries will implement various emission control strategies in response to climate change, such as carbon neutrality (Liu et al., 2021; Zhang et al., 2021), which will also alter the emission structure. How these extreme weather events and control measures influence the temporal characterization of sources, the formation of air pollution, and the spatial-temporal contribution of emissions from different countries, as well as their interactions, are also worth further investigation in the future. Such investigations can foster mutual cooperation among nations to collectively address environmental challenges.* "